



# Sea ice and water classification on dual-polarized Sentinel-1 imagery during melting season

Yu Zhang[1, 3], Tingting Zhu[2, 3], Gunnar Spreen[3], Christian Melsheimer[3], Marcus Huntemann[3], Nick Hughes[4], Shengkai Zhang[1], and Fei Li[1]

[1]Wuhan University, Chinese Antarctic Center of Surveying and Mapping, China

[2]Wuhan University, State Key Laboratory of Information Engineering in Surveying, Mapping and Remote Sensing, China

[3]University of Bremen, Institute of Environmental Physics, Germany

[4]Norwegian Meteorological Institute - Ice Service, Tromsø, Norway

**Correspondence:** Yu Zhang (yuzhang_spl@whu.edu.cn); Tingting Zhu (zhutingting62008@163.com)

**Abstract.** We provide a new sea ice and water classification product with high spatial and high temporal coverage using Sentinel-1 Synthetic Aperture Radar (SAR) data. The classification is applied in the Fram Strait region in the Arctic during melting seasons, when the contrast between backscatter intensities of different ice types observed by SAR is reduced due to the melted ice surface and wet snow on sea ice. The wet or melted snow strongly reduces the SAR penetration depth and thus suppresses the volume scattering contribution of sea ice. Furthermore, within the marginal sea ice zone (MIZ) ambiguities between ice and water can result from the effects of winds and ocean currents on the ocean SAR backscatter. On the other hand, under calm conditions the contrast between thin ice and flat open water can be reduced, and thus decrease the separability of some ice. In summary, the melting season represents the most challenging time of the year for reliable ice-water classification from SAR data. We propose here a new approach to overcome these problems by using a mixture statistical distribution based conditional random fields (MSTA-CRF) model. To obtain reliable ice-water classification whilst maintaining a fast computation time suitable for operational applications, the MSTA-CRF adopts a superpixel approach in the fully connected CRF model. The MSTA-CRF is a semantic model, which integrates statistical distributions (Gamma, Weibull, Alpha-Stable, etc.) to model the backscatters of ice and water and overcome the effects of speckle noise and wind-roughened water. Dual-polarization Extended Wide (EW) mode Sentinel-1A/1B SAR data with 40 m spatial resolution is available several times per day within the Fram Strait region. Observations from June to September during the six years 2015-2020 are collected and classified into ice and water categories. The classification performance of algorithm is evaluated using ice charts from the Ice Service at the Norwegian Meteorological Institute (MET Norway). The methods of training sample selection, and their application to processing large data volumes and automatic classification of ice-water are discussed. In the experiment part, we demonstrate that the MSTA-CRF can provide a good performance with about 90% accuracy for ice-water classification, which is better than most of other state-of-the art algorithms. Compared with the 89 GHz microwave radiometer ASI sea ice concentration product, the sea ice extent in Fram Strait derived from MSTA-CRF algorithm is lower during melting seasons from 2015 to 2020, and the monthly June to September sea ice area does not change so much in 2015-2017 and 2019-2020, but it has a significant decrease in 2018.

**Keywords:** sea ice; ice-water classification; statistical distribution; sea ice concentration; SAR; Sentinel-1; CRF; Fram Strait



## 1. Introduction

Synthetic Aperture Radar (SAR) is widely used in sea ice monitoring of sea ice concentration, area, leads, ice floe, ice edge, ice classification and deformation (Dierking, 2010). With its all-weather day-and-night, high spatial resolution and subsurface imaging capabilities, SAR has become the most important observation technique for operational sea ice monitoring related to maritime navigation and search-and rescue (Liu et al., 2015; Zakhvatkina et al., 2017). From the early 1990s, spaceborne C-band SAR such as ERS-1/-2, ENVISAT, RADARSAT-1/-2(RS1/RS2) and Sentinel-1A/-1B have been used as the primary data sources for the operational ice services since it provides good contrast between open water (OW) and ice categories (Johannessen et al., 2007). Sentinel-1 is the major data source for operational sea ice and iceberg monitoring in the polar regions from the European Commission's Copernicus programmer, and is freely available to registered users.

Ice charts are important for operational applications like ship navigation, and scientific studies such as the mass balance between atmosphere-ocean-ice systems. Earlier generations of SAR sensor provided only single-polarization HH or VV (horizontal or vertical transmit and receive polarization) where it can be difficult to distinguish sea ice from wind-roughened open water (Scheuchl et al., 2005). Newer SAR sensors introduced a cross polarization HV (horizontal transmit and vertical receive polarization), or the opposite VH, channel where the backscattering coefficients are relatively independent of the water surface roughness conditions caused by high wind speed (Dierking, 2013). This can be utilized to improve water detection as it provides good contrast between wind-roughened water and ice during freeze-up periods. However, the HV channel usually has a lower signal to noise ratio. In some cases, polarimetric features for improved distinguishing of different ice types are used when the full polarimetric data is available (Ressel et al., 2016; Johansson et al. 2017). The co/cross polarization (i.e. HH/HV or VV/VH; Komarov et al., 2021) is useful in sea ice classification since it not only provides a better contrast between ice and water, but also decreases the instability of backscatter value in open water areas caused by high speed wind. (Liu et al., 2015; Zhu et al., 2016; Zakhvatkina et al., 2017). The Map-Guide Sea Ice Classification System (MAGIC) integrates Iterative Region Growing using Semantic (IRGC) classification (Yu and Clausi, 2008) with pixel-based Supported Vector Machine (SVM) classifier. A classification accuracy of 96.5% with respect to manually drawn ice charts was achieved using a limited number (20) of RS-2 scenes in Beaufort Sea area (Clausi et al., 2010; Ochilov and Clausi, 2012; Leigh et al., 2014). Moreover, the operational Ice Service at the Norwegian Meteorological Institute (MET Norway) use high-resolution SAR data from Sentinel-1, RS-2 and COSMO SkyMed, in combination with optical imaging from Sentonel-2 and -3, NASA Suomi NPP VIIRS, NOAA AVHRR for manual interpretation, with low resolution passive microwave form low resolution microwave from AMSR2 used as a last resort if no other data is available.

Backscattering characteristics determined by sea ice surface roughness and its dielectric properties can be fully explored to separate different sea ice types and water. As the backscattering is usually affected by ocean waves propagating into the ice area, it is not enough to only rely on backscattering intensity in identifying the different ice types and water. Additional information such as statistical distribution of sea clutter performs well in distinguishing water and ice. For



modeling the backscatter signal in SAR imagery, the K-distribution is a good model for four-look HH-polarized SAR data from Arctic sea ice (Roberts and Furui, 2000; Joughin et al., 1993). Since scatters of the sea clutter cannot be completely uniformly distributed (Ward et al., 2006), the mixture of several statistical distributions is needed. Statistical distributions combined with advanced classifiers have been widely used to improve ice and water classification (Barber and Ledrew, 1991, Nystuen et al., 1992). Besides the statistical distribution-based methods, Artificial Neural Network (ANN) characterized by modeling nonlinear relationships between the input coherency matrix features and ice categories (Ressel et al., 2016) shows promising potential in ice water analysis. SVM realize ice water classification by training the kernel with the transformation into high dimensional space, which solves the problem of undistinguishable features in low dimensional space. Textual feature based neural network methods also shows good performance in ice-water classification, especially in the marginal ice zone (Aldenhoff et al., 2018). Murashkin et al. (2018) use textural features and a random forest classifier to detect leads within the ice pack from Sentinel-1 data but do not focus on th MIZ. Markov random field (MRF) has been widely used in sea ice segmentation and classification since MRF framework modeling the contextual information by utilizing the Gaussian statistics to exploit backscatter characteristics also shows its efficiency in sea ice classification (Clausi, 2001). However, MRF usually assumes that the observation vectors are conditionally independent, which ignores the interactions between ice and water, thus it is difficult to describe the dynamics of sea ice scenes especially in melting seasons. Conditional random fields (CRF) can capture the contextual information in both labels and observed data, and directly model the posterior as a Gibbs distribution and thus allow capturing the dependence of the observed data (Wang, 2016). The CRF model solve the maximum posterior inference over the defined superpixels obtained by the segmentation, in order to optimize the heterogeneity of the superpixels, and the pixel-wise ice maps obtained by SVM are utilized as the category prior to updating the label in superpixel (Zhu et al., 2016).

In this paper, the Mixture Statistical distribution based fully connected CRF (MSTA-CRF) framework is proposed for accurate and efficient ice-water classification. Our objective is to use this statistical information in a Bayesian classification scheme for an operational ice-water classification algorithm. The higher order potentials method not only considers the neighboring information but also the long-range interactions between pixels. Statistical distribution such as Rayleigh, Weibull, Gamma, Log-Normal and Alpha-Stable are integrated in the CRF framework, parameter estimation of mixture statistical distribution is investigated in this work. Considering the heterogeneity of the superpixel, a sub-superpixel based FCNN (Fully connected neural network) is used to represent the hidden information corresponding to the spatial and semantic relationship for each superpixel. We propose a mixture of statistical distributions based on fully connected CRF for ice-water classification using Sentinel-1 dual polarized SAR data with the following goals:

1. Roughened ice-free water caused by winds and ocean currents make it difficult to distinguish water from ice in the co-polarization (HH or VV) channel of SAR data. Statistical distribution is proposed for modeling the SAR backscatter signal to deal with the wind-roughened ice-free pixels. Then, we incorporate the statistical distributions into the proposed CRF framework which considers the spatial and semantic information.

2. During the melt seasons, sea ice deformation features shows much more textual that decrease the severability between flat thin ice and calm water. Most sea ice classification methods rely on multiple features to input, e.g.,

into SVM. To deal with the large computational cost, these approaches usually use a down-sampling strategy, which results in losing a lot of thin structures such as leads. We propose sub-superpixel, based MSTA-CRF models, to reduce processing time as well as improve the accuracy of ice-water classification.

The statistical distribution is used to reduce the influence of the wind roughened water. The use of superpixels can improve ice-water classification results by preserving leads and ice edges. An operational ice-water classification algorithm by MSTA-CRF model is proposed for addressing problems of ice-water classification during melt seasons in this paper.

This paper is organized as follows: In section 2, we give a brief description of the research area and data used. In section 3, we introduce the framework of the proposed algorithm, and give details of the methodology including pre-processing, training data selection, MSTA-CRF modeling and classification. We evaluate the ice-water classification results with the help of MET Norway ice charts to manually delineate the testing and training samples of open water and ice for MSTA-CRF in Section 4. In section 5, we discuss the dynamics of sea ice during melting seasons. After that, we give our conclusion in Section 6.

## 2. Data Source

### 2.1. Research area

To consider the spatial contextual information and preserve the spatial details of each pixel in SAR imagery, the energy function based maximum a posteriori (MAP) estimation in MSTA-CRF framework is proposed for operational ice water classification during melting seasons in Fram Strait. Fram Strait is between the northeast of Greenland and northwest of Svalbard, and is the main outflow passage for sea ice export of the Arctic Ocean thereby controlling the mass balance of the Arctic Ocean sea ice cover (Maslanik et al., 2009; Spreen et al., 2009, Ludwig, et al., 2020). Increasing ice exports of Fram Strait in summer will contribute directly to open water further north (Metfies et al., 2017; Spreen et al., 2020). Figure 1 shows an overview of the research area and some satellite scenes used in this manuscript.

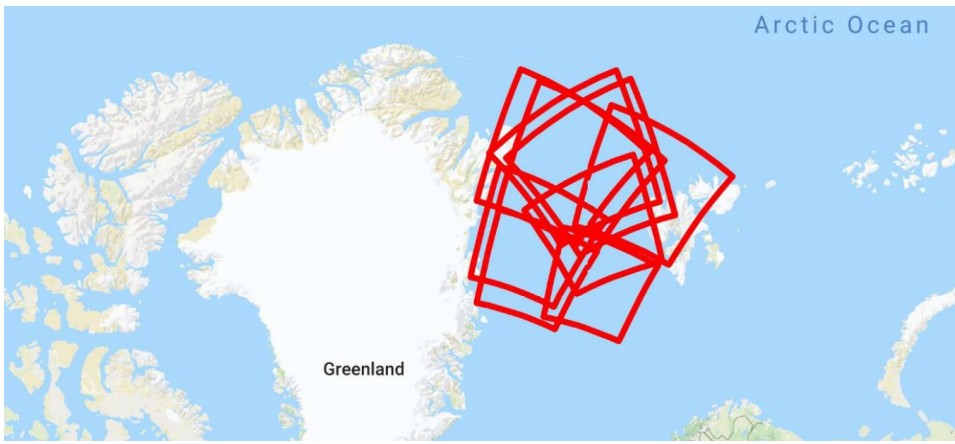

**Figure 1.** The location of the Sentinel-1 SAR scenes used for the experiment in this paper. The red rectangle illustrate some senses of Sentinel-1 SAR image, and the research area of Fram Strait defined in this paper is [75°N~83°N] [-15°W~15°E], the base map is cut from © Google Map.

### 2.2. Data source



**A) Sentinel-1 SAR data**

The two Sentinel-1 satellites are a part of the European Commission's Copernicus program. Sentinel-1A was launched on 3 April, 2014 and Sentinel-1B on 25, April, 2015 in a near-polar, sun-synchronous orbits. The Sentinel-1 constellation system can provide spatial resolution up to 5m (Stripmap mode) and high temporal resolution with a repeat time of 6 days

in four different operative modes: stripmap (SM), wave (WV), interferometric wide swath (IW), and extra wide swath (EW). For ice-water classification in this paper, the EW model Ground Range Detected (GRD) data during melting seasons from 2015 to 2020 are used. The details are listed in Table 1.

**Table 1.** Parameters of Sentinel-1 data.

| Acquiring Mode | Extra Wide Swath (EW) | | | | |
|---|---|---|---|---|---|
| Product type | Ground Range Detected (GRD) | | | | |
| Polarization | HH+HV | | | | |
| Ground Range Coverage (km) | 410 | | | | |
| Parameters of Different Sub-Swath | EW1 | EW2 | EW3 | EW4 | EW5 |
| Resolution: Range x Azimuth (m) | 90.9 x 90.1 | 93.1* x 89.4 | 93.3 x 86.9 | 93.8 x 86.5 | 95.1 x 90.1 |
| Pixel spacing: Range x Azimuth (m) | 40 x 40 | 40 x 40 | 40 x 40 | 40 x 40 | 40 x 40 |
| Equivalent Number of Looks (ENL) | 15.2 | 9.7 | 9.6 | 9.5 | 9.6 |
| Incident Angle (°) | 23.7 | 30.9 | 36.2 | 40.9 | 44.5 |

**b) MET sea ice chart**

The operational Ice Service at Norwegian Meteorological Institute (MET Norway) uses high-resolution SAR data and optical data from Sentinel-2 and 3, NASA Suomi VIIRS, NOAA AVHRR for manual interpretation, with low resolution passive microwave from AMSR-2 used as last resort if no other data is available for manual interpretation by its ice analysts. The sea ice concentration classes, including whether the ice is drifting or (land) fast, together with reference colors are shown in Table 2. For the ice-water classification in this study, the Ice Free, Open Water and Very Open Drift Ice are

defined as water categories with the sea ice concentration (SIC) less than 1 (tenths level), and Open Drift Ice, Close Drift Ice, Very Close Drift Ice and Fast Ice are defined as the ice categories. The accuracy of different algorithms is calculated pixel-by-pixel with the MET Norway ice charts. The validation of sea ice classification using MET Norway ice chart will be discussed in Section 4.

**Table 2.** Ice type and reference color of MET Norway ice chart

| Ice type | SIC in tenths | Centre SIC % | Color |
|---|---|---|---|
| Ice Free | 0 | 0 | 🔵 |
| Open Water | 0-1 | 5 | 🔵 |
| Very open Drift Ice | 1-4 | 25 | 🔵 |
| Open Drift Ice | 4-7 | 55 | 🔴 |
| Close Drift Ice | 7-9 | 80 | 🔴 |
| Very Close Drift Ice | 9-10 | 95 | 🔴 |
| Fast Ice | 10 | 100 | 🔴 |

**c) ASI sea ice concentration**

The daily ASI sea ice concentration (Spreen et al., 2008) product from AMSR2 is also used to validate the performance of MSTA-CRF algorithm. The ASI product used in the manuscript is retrieved using the 89 GHz passive microwave dataset with the spatial resolution of 3.125km, and it is downloaded from the following website: www.seaice.uni-bremen.de.



## 3. Methodology

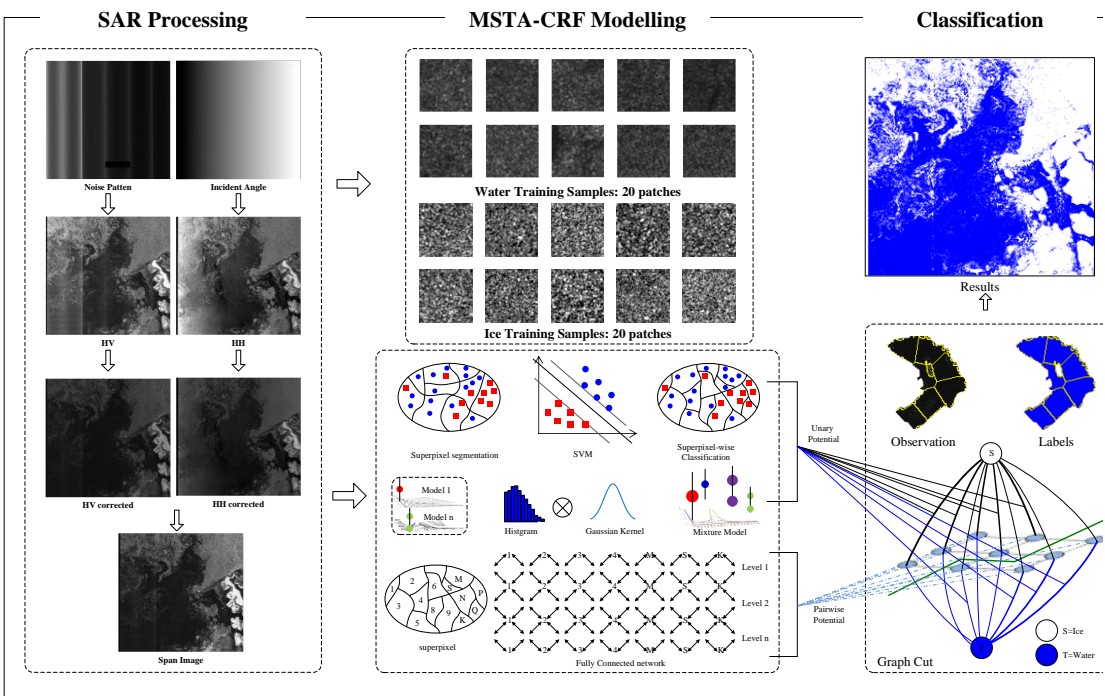

**Figure 2.** The framework of the proposed MSTA-CRF algorithm for ice-water classification.

We use the following framework for supervised sea ice classification from Sentinel-1 SAR data. Here we give a brief summary of the three involved steps (Figure 2), while the details will be explained in the following sections.

(I) Preprocessing: we apply a thermal noise removal and incident angle normalization on the dual polarization Sentinel-1 SAR images, before calculating the SPAN of the HH and HV channels ($\sqrt{HH^2 + HV^2}$). The SPAN takes the advantage of the two channels and is used in all following steps.

(II) Prior to classification we model the SAR data by combining a Mixture of three STAtistical (MSTA) distributions (Log-Normal, Rayleigh, and Alpha-Stable) with conditional random field (CRF) theory (MSTA-CRF). To reduce computation time we follow a hierarchical approach: the SAR images are first divided into several roughly homogeneous segments called superpixels based on mean shift theory (>5000 pixels at $40 \times 40$ m$^2$). Then we use another, more restricted, mean shift procedure to segment the superpixel into even more homogeneous sub-superpixels (the mean size of the sub-superpixels contains 24 pixels, 0.038km$^2$). These sub-superpixels are the smallest entity our ice-water classification is based on.

(III) The sub-superpixel classification uses three weighted decision criteria referred to as potentials within the CRF context: The first potential uses a support vector machine (SVM) classification on all pixel in a sub-superpixel based on their backscatter. As second potential, the similarity of the previously fitted MSTA backscatter distributions for the ice and water class to the backscatter distribution found in the sub-superpixel is used. The third potential describes the heterogeneity of the superpixel by using a fully connected network on the relationship of different sub-superpixels which





accounts for their location and backscatter similarity. During training, we use square training samples as superpixels for each category (Figure 2 top-middle) and segment them into sub-superpixel. Then the parameters of the three potentials described above as well as their corresponding weights will be trained to generate the MSTA-CRF classifier for sea ice and water classification.

Finally, we use a graph cut classifier based on these three potentials to maximize the probability of the MSTA-CRF model. As a result each sub-superpixel is labeled with either the ice or water category. The details of MSTA-CRF including the above mentioned steps will be described in the following sections.

### 3.1. Pre-processing of SAR data: incidence angle normalization and noise reduction

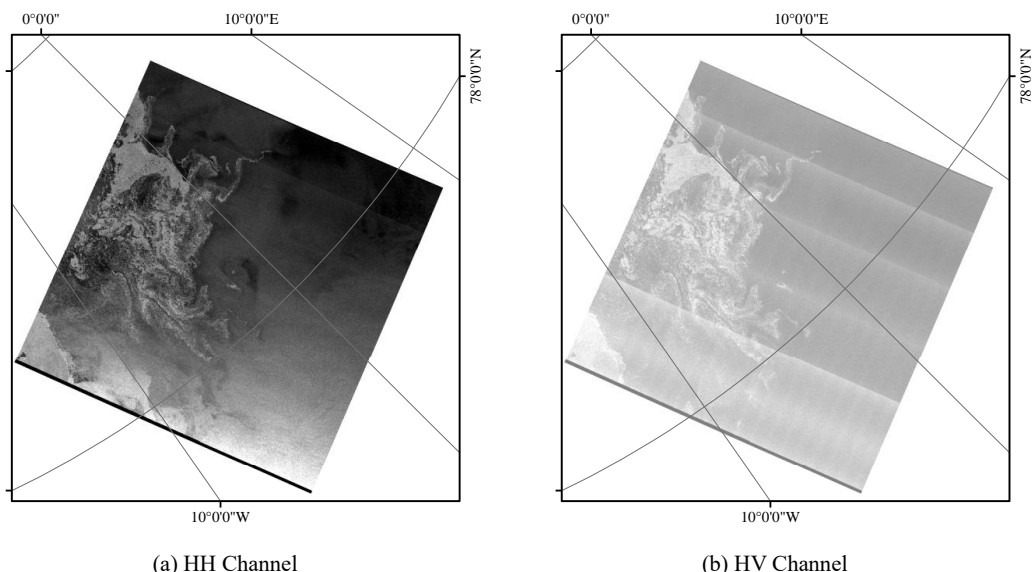

(a) HH Channel                    (b) HV Channel

**Figure 3.** Example image of Sentinel-1 EW dual-polarization data in Fram Strait on 2015-08-25, (a) HH channel with angular dependence, (b) HV channel with thermal noise.

For C-band dual-polarization Sentinel-1 data, the backscatter coefficients at different incidence angles can vary

significantly from the near to far range. For the HH channel of sample image in Figure 3(a), backscattering coefficients at near range is much larger than that at far range. In the HV channel, the dependence of backscatter on the incidence angle is not that obvious as it also contains bias of backscattering coefficient for different sub-swath. In addition to the angular dependence in the HH channel, contrast between open water and ice is much lower compared to the HV channel (Figure 3(b)). Ice-water classification along marginal ice zone can be significantly influenced by winds and ocean waves due to

wind-roughened water surface leading to much larger backscattering values. However, the thermal noise including the scalloping noise and noise floor in the HV data limited its application for automatic analysis algorithm (Park et al., 2018). To reduce the effect of incidence angle, radiation calibration is required especially for the HH channel. Noise reduction is required for HV data (Figure 3(b)). Preprocessing methods including incidence angle normalization and noise reduction methods are performed as described below.

The backscattering power from sea ice and open water is determined by several factors including orientation, slope,





roughness, surface cover and permittivity. The combination of these factors may lead to additional heterogeneity in vertical or horizontal directions, which can further impact the intensity and amplitude of the backscattering signal. The radar angular configuration, polarization, and frequency are the major instrument-related parameters. The radiometric angular-dependence is corrected by a cosine law (Gauthier et al., 1998; Mladenova et al., 2013) using the local incidence angle.

This approach assumes that the amount of power that is backscattered towards the sensor direction is proportional to the path length and thus to the cosine of the incidence angle, Since the radiation variance as a function of the observed area is also cosine function, the measured backscatter is described as:

$$\sigma_{\theta_i}^0 = \sigma_0^0 cos^n(\theta_i) \tag{1}$$

$$\sigma_{ref}^0 = \frac{\sigma_{\theta_i}^0 cos^n(\theta_{ref})}{cos^n(\theta_i)} \tag{2}$$

Where the measured backscatter $\sigma_{\theta_i}^0$ is related to the sea ice surface roughness dependence parameter n and it can be derived from the slope of the measure backscatter $\sigma_{\theta_i}^0$ and the local incidence angle $\theta_i$ by the linear regression model.

$\theta_{ref}$ and $\sigma_0^0$ are the reference incidence angle and nadir backscatter respectively. The normalized backscatter $\sigma_{ref}^0$ is defined by calculating the ratio of measured backscatter $\sigma_{\theta_i}^0$ at the reference angle of $\theta_{ref}$ and incidence angle with a cosine function. This normalization method provides the observation at a constant incidence angle according to the application. To deal with the SAR images acquired in different conditions and compare the backscatter values of sea ice in the reference angles, absolute radiation calibration is carried out to convert the amplitude value of HH channel to $\sigma_0$. Then

the incidence angle normalization is utilized to transform the backscattering value into the reference angle of 23°, the details of reference angle selection will be discussed in the experiment part in Section 4.1.

Due to the sparse ground control points in the XML file (121 points for nearly 400 km swath) of EW mode Sentinel-1 data, the noise floor of the neighboring sub-swaths in HV channel is discontinuous. The noise level of Sentinel-1 data before September 2016 and afterwards are different. The noise reduction mainly relies on the XML file provided by

Sentinel-1 head file data and derived by the interpolation of the noise tie points for each pixel. Then the corrected SAR backscatter is calculated by the subtraction of the backscatter values of the SAR and the noise pattern. $\sigma_{full}^0$ is the sigma naught in the look up table.

$$\sigma^0 = (DN^2 - noise_{full})/(\sigma_{full}^0)^2 \tag{3}$$

The subtraction of the noise of the entire image for each pixel from the digital number (DN) is divided by the backscatter coefficients provided in the XML file. With respect to the first sub-swath of each SAR scene, FFT transform

is utilized to remove the remaining scalloping noise with high frequency (the amplitude of FFT transform of high frequency is set to zero) (Choi et al., 2019). To prevent the yielding of noise floor of the neighboring sub-swath, a 10-pixel wide stripe of data along the edge of each sub-swath is not taken into consideration for the further analysis.

Figure 4 illustrates the incidence angle correction and thermal noise removal results on two sample images. For better illustration, the contrast of the image is enhanced. During the melting seasons, the ice-covered areas, and the rough OW

areas appear both bright in the HH channel and are therefore difficult to distinguish, which can be seen in Figure 4 (a). For HV channel in Figure 4(c), the thermal noise caused by the uncertainty of noise floor between different sub-swaths can





also result in misclassification. Both HH and HV channels have been corrected as shown in Figure 4 (b) and (d). Angular dependence is corrected using the reference angle of 23°. The incidence angle correction will inevitably lead to the loss of some information and the contrast of the corrected image is decreased compared with the original image. Nevertheless, it shows a high distinguishability between sea ice and open water. For HV channel, the thermal noise caused by the discontinuity of noise floor of the neighboring sub-swath at far ranges have been removed. For the near range (first and the second sub-swath), the overall image quality is greatly improved, although there is a small amount of noises remaining.

In the corrected HH channel, backscattering coefficients over wind-roughened water surface make it difficult to be distinguished from thin ice along marginal ice zone. In corrected HV, backscatter coefficients of thin ice are greatly reduced. The Span image takes advantage of both HH and HV channel, with SPAN being defined as the total power calculated by the square root of HH and HV. The equation for SPAN is given as below, $\sigma_{HH}$ and $\sigma_{HV}$ are backscatter value of HH and HV channel respectively.

$$\sigma_{SPAN} = \sqrt{\sigma_{HH}^2 + \sigma_{HV}^2} \tag{4}$$

It is clear to see that the rough area in the near range caused by the wind and ocean current is removed as shown in Figure 5 (e). Span data as combination of the HH and HV channels reduces the amount of data and thus is helpful for ice-water classification. We will used the Span in the following for all classifications.

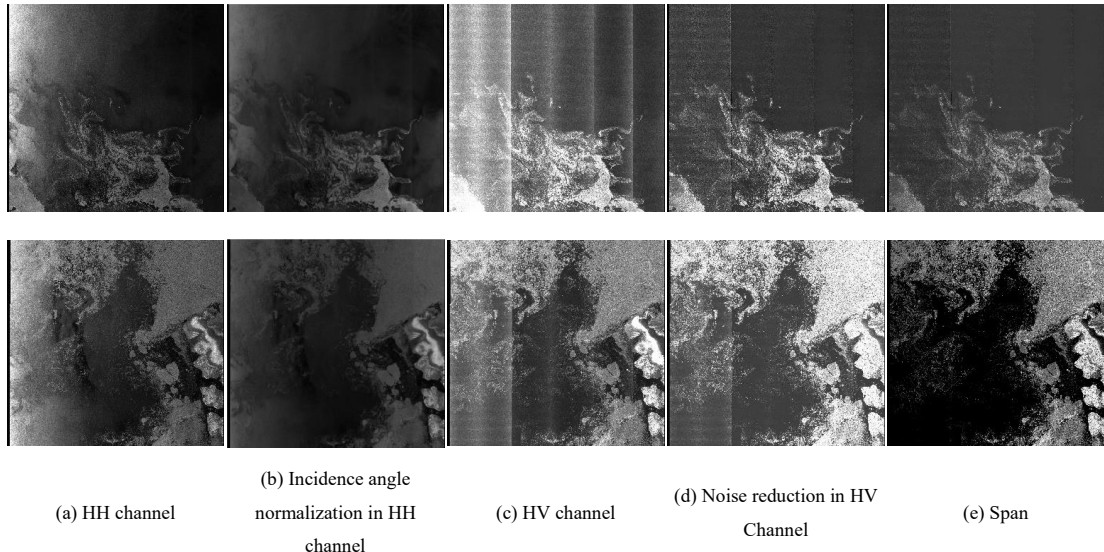

(a) HH channel | (b) Incidence angle normalization in HH channel | (c) HV channel | (d) Noise reduction in HV Channel | (e) Span

**Figure 4.** Preprocessing of sample images: upper (2015/08/25), bottom (2017/08/18). (a) and (c) are the original HH and HV channels, (b) and (d) are the incidence angle normalization of HH channel and noise removal for HV channel respectively, (e) is the span image calculated by $\sqrt{HH^2 + HV^2}$.

### 3.2 Training samples selection

The training samples for sea ice and water classification is selected from the Sentinel-1 data acquired from 2015 to 2018 for the Sentinel-1 SAR imagery in the Fram Strait area. We first select one image for each day from 2015 to 2018 during June to September (488 images in total). Then 10 patches for each category (i.e., ice and water) are selected from the image dataset randomly using the MET Norway ice chart, and the size for each patch is 64*64 pixel. As a result, we





got 9760 patches (4880 for ice and 4880for water), which is called the training dataset.

During the training step, we first select 100 patches for each categories to generate the MSTA-CRF model, each patch of the training samples is labeled pixel by pixel. This, e.g., means we know where the ice sub-superpixel is for an otheraise roughly homogenous water training sample. For the statistical distributions training the contribution of the ice sub-

superpixel for the probability of the corresponding water superpixel is set to 0, while for the fully pairwise potential, the ice sub-superpixel will be used to calculate the pairwise potential with the other sub-superpixels. The performance of the classifier is validated on all the training datesets. If the overall accuracy (OA) is lower than 99%, we add 100 patches (50 for ice and 50 for water) from the rest of the training dataset to train the revised model, and it will also be tested on the whole training dataset. For the sea ice classification in this paper, we repeat the training procedure for ten times and find

that when the training samples reaches 100, the OA is over 99%, which means that 1000 patches are enough to get a satisfied MSAT-CRF model, and it only occupies 10% of the full training dataset.

### 3.3 Mixture statistical distribution based fully connected CRFs

CRF (Conditional Random Fields) is a framework for constructing probabilistic models, which have been widely used in segmentation and classification (e.g., Tuia et al., 2018). For the operational ice-water classification, we define CRF on

superpixels instead of performing a pixelwise classification. A superpixel is a group of homogeneous pixels that rendered with uniform backscatter coefficients. These superpixels are obtained by the unsupervised mean-shift algorithm (Comaniciu and Meer, 2002). Generation of accurate superpixels is difficult, especially thin structures, such as leads are misclassified. We divide the superpixels into smaller sets as sub-superpixels $p\epsilon S$ using mean shift method. The sub-superpixels are now the smallest units we consider for the statistical distribution calculation. Then, we have the mixture

statistical distribution based fully connected CRF as the form:

$$P(x|y) = \frac{1}{Z_{MSTA-CRF}} exp\left[\underbrace{\sum_{p\epsilon Ss}\omega_1 f_p^{SVM}(x_i|y_i) + \sum_{p\epsilon S}\omega_2 logP(y_i|x_p)}_{unary\ potential} + \underbrace{\sum_{p,q\epsilon S}\sum_{q\neq p}\omega_3 f_{pq}(x_p,x_q|y_i)}_{fully\ connected\ potential}\right], \quad (5)$$

where $Z_{MSTA-CRF}$ is the partition function. and $\omega_n$ are weight parameters of the MSTA-CRF model.

The unary potential is defined on the individual sub-superpixels at site $p$. $y_i$ accounts for SAR backscatter coefficients pixel $i$ and $x_i$ denotes its category (ice or water). $f_p^{SVM}(x_i|y_i)$ deals with the single pixels individually and is represented as

$$f_i^{SVM}(x_i|y_i) = \sum_{i\epsilon s}\delta(x_i,l)logP(x_i|y_i), \quad (6)$$

where $\delta(x_i,l)$ is the Dirac function, where $\delta(x_i,l) = 1$ for $x_i = l$ for the ice type, and $\delta(x_i,l) = 0$ for $x_i \neq l$ with open water category. The local conditional distribution $P(x_i|y)$ is obtained via Support Vector Machine (SVM). $logP(y_i|x_p)$ is calculated by statistical distribution models and the fully connected potential, which will be given in following section.

### 3.3.1 Mixture statistical distribution

The conditions of the scatters in a resolution cell can be treated as the sum of elementary scattering areas on a rough surface due to its random walk characteristics in the complex plane (Goodman, 1976). The statistical model for speckle noise is on the assumption that each pixel contains a great number of scatters of radiation with a wavelength of C-band





that is comparable to the roughness of the sea ice surface. For the lack of detailed information on the microscopic structure of the surface, the statistical attributions of the speckle patterns seem to be a better way to understand the characteristics of the radar signal. Backscatter signals of SAR image originally forms by the random scatters in the radar backscattering, and can be modeled by statistical distribution models, e.g. Rayleigh, Gamma, Log-normal, Weibull and Alpha-stable,

which have been exploited to model the heavy-tailed and sharp-peaked statistical properties of SAR imagery under complicated coherent noise condition (Yijing, et al., 2013).

We define the statistical potential $logP(y|x_p)$ on sub-superpixels and model each superpixel with different statistical distributions. The parameter estimation methods for each statistical distribution are presented in Table3.

**Table 3.** Single statistical distribution model and parameters estimation method used in this paper.

| Statistical distribution model | Parameters | Method of Logarithmic Cumulants |
|---|---|---|
| Rayleigh | $\sigma$ | $k_1 = (ln2 + \psi(1))/2 + ln\sigma$ |
| Gamma | $\mu, L$ | $k_1 = \psi(L) - lnL + ln\mu \quad k_2 = \psi(L)$ |
| Log-Normal | $\mu, \sigma$ | $k_1 = \mu \quad k_2 = \sigma^2$ |
| Weibull | $\mu, \eta$ | $k_1 = ln\mu + \eta^{-1}\Psi(0,1) \quad k_2 = \eta^{-2}\Psi(1,1)$ |

Parameters $\{\mu, \sigma, b, c\}$ are estimated by method of logarithmic cumulants (MoLC) (Vladimir, 2013). In MoLC, Melin

transformation method is used to represent probability density function (*pdf*) (Cui, 2012).

For each single statistical distribution, logarithmic cumulants k can be used to estimate the corresponding distribution parameters as shown in Table 3. The PDF reconstruction for each distribution is shown in Section 4.3.

With respect to the alpha-stable distribution without analytical expression, we present it as follows:

$$q(\varphi) = \begin{cases} \exp\left\{j\mu\varphi - |\gamma\varphi|^\alpha \left[1 - jsign(\varphi)\beta \tan\left(\frac{\pi\alpha}{2}\right)\right]\right\}, \alpha \neq 1 \\ \exp\left\{j\mu\varphi - |\gamma\varphi| \left[1 + jsign(\varphi)\beta \frac{2}{\pi}\ln(|\varphi|)\right]\right\}, \alpha = 1 \end{cases} \quad (7)$$

where sign is the sign function. $\{\alpha, \beta, \gamma, \mu\}$ are the parameters of the alpha-stable model. $\alpha$ is the characteristic

exponent and $\beta$ is the skewed parameter. $\gamma$ is the dispersion parameter and $\mu$ stands for the location parameter. The distribution parameters of the alpha-stable distribution are estimated by using the simulated annealing method (Grosse, 2007; Geman 1984).

With the increasing spatial resolution of SAR, image scene becomes more and more complex, to deal the SAR image with small inter-class differences, a mixture statistical distribution is integrated into the CRF framework to discriminate

the ice category from open water using Sentinel-1 SAR images. After estimating parameters of a single statistical distribution, we can formulate these statistical distributions in a weighted way as

$$\sum_{i=1}^{M} lnF(y_m, \theta) = \sum_{i=1}^{M} ln[\alpha_i f_i(y_m, \gamma_i)] \quad (8)$$

The weighting parameter $\theta \sim (\alpha_i, \gamma_i), i = 1, ..., M$ is estimated using an expectation–maximization (EM) algorithm (Saldju, 2000).

### 3.3.2 Fully connected CRFs

The unary potential of the CRF model can capture the dependence of the observed data. For each sub-superpixels, the

fully connected network is constructed to represent the hidden information corresponding to the spatial and semantic relationship.

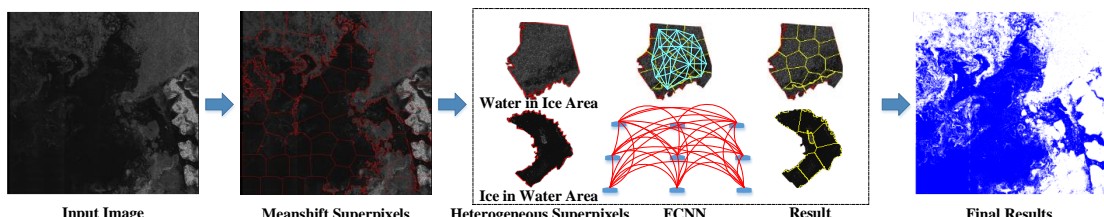

Input Image    Meanshift Superpixels    Heterogeneous Superpixels    FCNN    Result    Final Results

**Figure 5.** Illustration of fully connected network of superpixel. The span image is first divided into superpixel using mean-shift method. For each superpixel, it is treated as the heterogeneous area and will be divided into sub-superpixel under more restrict mean-shift procedure. The relationship of different sub-superpixels which accounts for their location and backscatter similarity will be calculated in the fully connected network. Then we give the labels for each sub-superpixel and finally obtained the classification results.

For maintaining the accuracy of statistical distribution parameter estimation in the fully connected CRFs, the size of each superpixel is larger than 5,000 pixels. Each superpixel has a label to represent the dominating corresponding category of ice or OW. Since, however, there are still heterogeneous area within a certain superpixel, it will lead to a significant misclassification in the results. Thus, each superpixel is divided into several sub-superpixel using a random number (less than 50), then we can get a much more homogeneous area within a certain superpixel by integrating the Gaussian potential in the fully connected framework. It may have different labels to represent details with in the superpixel, but for the calculation of the pairwise potential between different superpixels, they are treated as a homogeneous area as usual.

Local fully connected potential of CRF models $f_{ij}^{pairwise}(x_i, x_j | y)$ imposes the spatial interaction and is defined as

$$f_i^{fully}(x_i, x_j) = \mu(x_i, x_j) \sum_{m=1}^{K} \omega^{(m)} k^{(m)}(y_i, y_j), \qquad i, j \in [1, N]. \tag{9}$$

where $\mu(x_i, x_j)$ is a simple label compatibility function. $y_i$ and $y_j$ are the SAR backscatter coefficients at a pair of sub-superpixel $i$ and $j$. $\omega^{(m)}$ are linear combination weights. Gaussian kernel $k^{(m)}(y_i, y_j)$ is defined as

$$k(y_i, y_j) = \omega^{(1)} \underbrace{\exp\left(-\frac{|p_i - p_j|^2}{2\theta_a^2} - \frac{|y_i - y_j|^2}{2\theta_b^2}\right)}_{appearance\ kernel} + \omega^{(2)} \underbrace{\exp\left(-\frac{|p_i - p_j|^2}{2\theta_c^2}\right)}_{smoothness\ kernel} \tag{10}$$

$p_i$ and $y_i$ are the positions and the backscatter coefficients for each sub-superpixel respectively. The Gaussian kernel $k^{(m)}(y_i, y_j)$ includes an appearance kernel that favors similar backscatter coefficients pixels and a smoothness kernel that removes small isolated ice floes. When nearby pixels are assigned different labels, a penalty term $\mu(x_i, x_j)$ with smoothness kernel is introduced and they can be calculated from the Potts model (Krahenbuhl and Koltun, 2012). The parameters $\theta_a$ and $\theta_b$ in the appearance kernel control the similarity and they can be learned from the training dataset.

### 3.4 Classification

In the CRF model, the final ice water classification is a maximization problem by assigning a label to each pixel of posterior probability. The global optimization is solved using the Graph cuts (Boykov et al., 2001) algorithm. As shown in Figure 2, a graph model is induced by defining the cut to separate the vertices (s, ice) as target and the vertices (t, open


water) as background. Each vertex corresponds to a sub-superpixel segment and they are connected by edges with different weight probability, and the ensemble of the edge is named a "cut". Then, the minimum cut problem is to use the least cost of a cut among all cuts separating the vertices, which is equal to the lowest sum of edge weights. The s-t edge weights minimum problem is converted into solving the minimum of the energy function including unary potential $f^{SVM}(L)$, the statistical distribution potential $f^{sta}(L)$ and the fully connected pairwise potential $f^{pairwise}(L)$. It can be expressed as:

$$E(L) = f^{unary}(L) + f^{pairwise}(L) + f^{sta}(L), \tag{11}$$

The inference procedure using the graph cut algorithm is exploited for indicating the label preference. We construct the graph with SVM potential, pairwise potential and the statistical distribution potential, shown in Figure 2. A S/T cut C on a graph with ice and water categories is used to partition the SAR image into two subsets, where S indicate ice label and T indicate water label. For the graph model, the cost of S/T nodes to the pixels is represented as a partial posterior marginal probability, which is derived from the unary potential and the statistical distribution potential in combination. Pixel interaction cost is represented as a joint posterior marginal probability, which is derived from the fully connected pairwise potential. The optimization goal is to find the minimum cost of a cut $C = \{S, T\}$, which is a sum of all the costs from partial posterior marginal probabilities and joint posterior marginal probabilities. Then, the minimum cost of label S/T indicates the sub-superpixel to be given either the ice or water label.

In this study, the pre-processing of Sentinel-1 imagery including incidence angle correction and thermal noise removal is performed by using Python functions. The MSTA-CRF classification algorithm as well as the comparison approaches are running in MATLAB 2016b on an Intel Core CPU i7 at2.8 GHz and 16GB of RAM. On this platform classification of one Sentinel-1 scene takes about 2000s.

## 4    Experiment and analysis

### 4.1. Selection of reference incidence angle

To choose the best reference angle for calculating the incidence angle corrected $\sigma^0$, we utilized the MSTA-CRF to calculate the OA (overall accuracy) and STD on the example images in Figure 3. The ice water classification results using difference reference angle ranges from 20° to 40° are shown in Figure 6. We use CV (coefficient of variance) to evaluate the performance of different parameters (Table 4). A lower CV means classification result are more stable. For the reference angle of 21°, 23°, 26°, 29°, 30°, 35°, 36°, the averaged OA are larger than 90%. The reference angle of 26° gets the largest OA but the STD is larger than 10%. As a tradeoff, the CV with the reference angle of 23° is chosen and used for the further sea ice water classification tasks.

**Table 4.** Evaluation of sea ice-water classification using different reference incidence angles

| | Reference incidence angle [°] | | | | | | | | | | | | | | | | | | | | |
|---|---|---|---|---|---|---|---|---|---|---|---|---|---|---|---|---|---|---|---|---|---|
| | 20 | 21 | 22 | 23 | 24 | 25 | 26 | 27 | 28 | 29 | 30 | 31 | 32 | 33 | 34 | 35 | 36 | 37 | 38 | 39 | 40 |
| OA/% | 88.75 | 91.11 | 89.74 | **91.78** | 86.55 | 89.93 | 92.78 | 86.55 | 85.76 | 90.33 | 91.24 | 84.27 | 85.73 | 82.44 | 87.67 | 90.03 | 90.46 | 87.22 | 83.28 | 86.26 | 87.35 |
| STD/% | 5.22 | 7.45 | 8.73 | **4.68** | 6.72 | 7.22 | 10.22 | 7.48 | 6.22 | 7.55 | 5.26 | 4.87 | 6.75 | 5.26 | 4.51 | 5.72 | 8.33 | 6.03 | 5.99 | 8.74 | 8.68 |
| CV | 0.059 | 0.082 | 0.097 | **0.051** | 0.078 | 0.080 | 0.110 | 0.086 | 0.073 | 0.084 | 0.058 | 0.058 | 0.079 | 0.064 | 0.051 | 0.064 | 0.092 | 0.069 | 0.072 | 0.101 | 0.099 |

### 4.2. Training samples selection for operational ice water classification

For ice-water classification, training samples are needed to train the MSTA-CRF classifier. It is, however, usually difficult to select in-situ measurements as the training samples because they do not cover a significant amount of the satellite scene. For operational application, the method also needs to be stable in time and for different regions without the need for re-training. To deal with the problem of sample selection for ice-water classification, the stability of MSTA-CRF



model using different training samples, as well as the performance of ice-water classification using a fixed model should be discussed. As a result, we would like to test whether the choice of different training samples may influence the final ice-water classification result. We also test which of the five statistical distribution are best suited for the MSTA-CRF and will keep only the significant ones.

In MSTA-CRF model, training samples mainly affect the calculation of statistical distribution parameters and the corresponding weight coefficients. As a result, the dynamic of these parameters using different samples is a direct indicator to represent the stability of the adapted model. In this paper, we randomly select 82 models from 2015-2107. The sensitivity of parameter using different training samples is shown in Figure 6. It is clear to see that $Gamma_L$ and $Weibull_\eta$ distributions are sensitive in some cases in Figure 6(a), but the corresponding weight in Figure 6(b) are only

slightly above 0. Thus they have a limited influence on the statistical distribution potential. $Log - Normal$ is not sensitive to the change of training samples, but its weight is just around 0.1, so it contributes a small part in the mixture model. Moreover, Alpha-stable  andRayleigh  are stable in most cases and the corresponding weight ranges from 0.3-0.4, which indicates that MSTA-CRF is not sensitive with the change of training samples and it mainly depends on Alpha-Stable, Log-Normal and Rayleigh model. We select these three distributions for our final classifier and only use them in the

following.

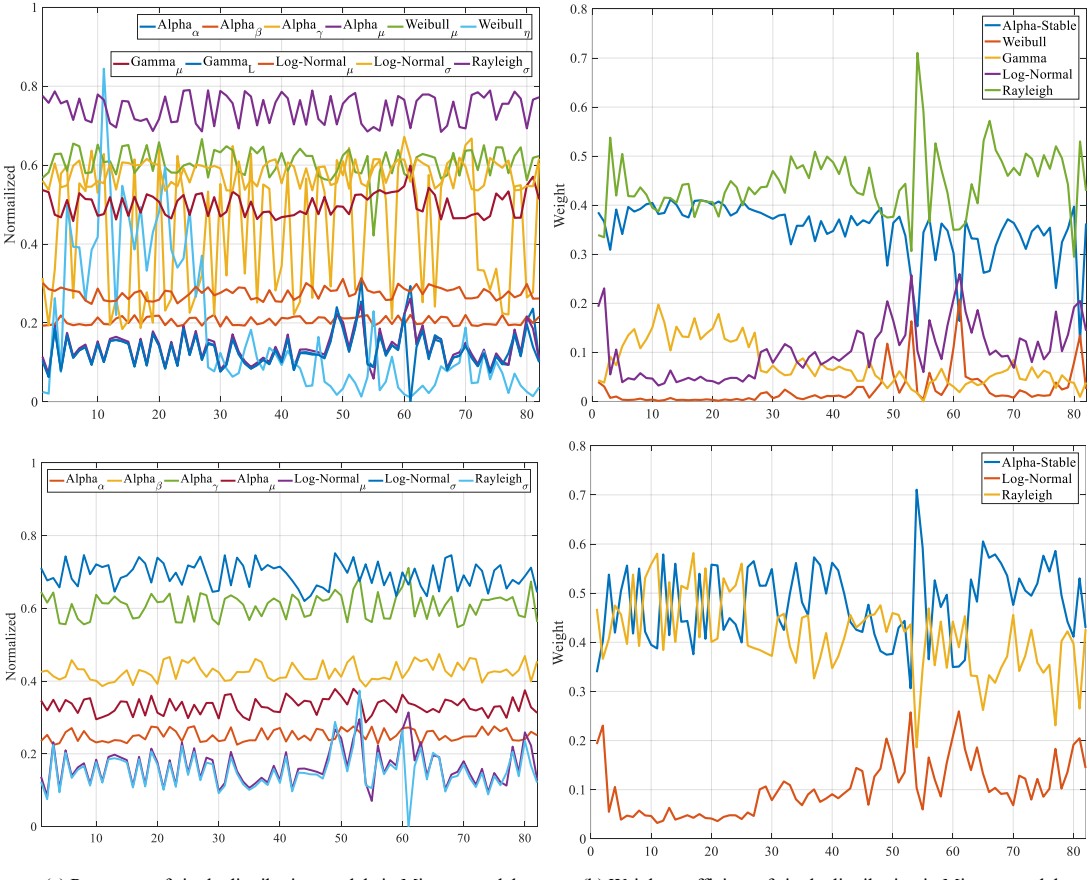

(a) Parameter of single distribution models in Mixture model        (b) Weight coefficient of single distribution in Mixture model



**Figure 6.** Sensitivity of parameters for the five statistical distribution models using 82 different SAR images (x axis) with different training samples, Alpha in (a) and (c) represents Alpha-Stable model.

The next step is the comparison of ice-water classification using a fixed model and an adapted model. The adapted model means the training samples are selected from the given test data using the sea ice chart. In this study, 20 training samples for each category are used for training the MSTA-CRF, where 10 of them are selected from the same month, 5 from same year but not from the same month, the remaining 5 from a different year and month. The fixed model means training samples are selected randomly from all the training dataset as described in Section 3.2. To evaluate the performance using fixed model and adapted model, 30 SAR images from 2015-2017 are used. The performance of these two models is shown in Table 5.

**Table 5.** Classification results using fixed model and adapted model

| Model type | Average Accuracy/% | STD/% |
|---|---|---|
| Fixed model | 88.02 | 6.87 |
| Adapted model | 89.51 | 5.81 |

Table 5 shows the average accuracy and STD using fixed model and adapted model respectively. The average accuracy using adapted model is slightly higher than for the fixed model since it contains the training samples from the test data, and the STD of the adapted model is lower than for the fixed model. For operational ice-water classification, however, the main task is to obtain the ice type and its distribution in unknown scenes. The performance of fixed model is only slightly lower than the adapted model with 1.5% lower in average accuracy and 1.06% higher in STD. The results indicate that the MSTA-CRF is not sensitive to the change of training samples and thus it can be used as an operational method for ice-water classification.

### 4.3. Comparison of different statistical distribution models

To evaluate the performance of different statistical distribution models, the reconstructed PDF (probability density function) and CDF (Cumulative Distribution Function) are compared with original SAR data for both ice and OW categories (Figure 7). The histograms for water (OW) and ice (bar plots) in Figure 7(a-f) are obtained by selecting 20 samples of each categories from the training dataset described in Section 3.2. For the PDF fitting curves in Figure 7(a)-(f), Rayleigh, Log-Normal, Alpha-Stable and the mixture model can get good correspondence for the ice histogram. For OW, all the five single distribution based models (a-e) underestimated the OW peak. The mixture model is the only one that can reconstruct the OW peak and shows the best agreement for both classes with an RMSD of about 0.03 (Table 6). From the CDF curves in Figure 7(h)-(i), we can see that the mixture model is most similar to the original SAR (followed by the Log-Normal distribution) data which means that the mixture model has a better performance than single distribution model. The comparison of the Root Mean Square Deviation (RMSD) for the different distributions with the manually classified SAR backscatter (test data) is given in Table 6. The RMSD for the Ice class using Alpha-Stable, Log-Normal and mixture model are similar. For the OW class the RMSD using mixture model is the smallest compared with the other models. Thus we conclude that using a mixture model is beneficial for the ice-water classification as especially the backscatter of the water class can be much better modeled compared to a single distribution model. Since the PDF reconstruction of Gamma and Weibull are not satisfying, we do not use them in the mixture model (see also next section). Thus also the corresponding




statistical distribution based CRF algorithm using these two models are not discussed in the following part.

5. **Table 6.** RMSD between PDF of the normalized backscattering coefficients from the test dataset and the different statistical distributions

|  | Gamma | Weibull | Log-Normal | Rayleigh | Alpha-Stable | Mixture |
|---|---|---|---|---|---|---|
| Sea Ice | 0.11 | 0.181 | 0.031 | 0.132 | 0.031 | 0.031 |
| Water | 0.094 | 0.176 | 0.071 | 0.113 | 0.073 | 0.026 |

.



(a) Gamma    (b) Weibull    (c) Log-Normal

(d) Rayleigh    (e) Alpha-Stable    (f) Mixture

(g) CDF of Ice    (h) CDF of Water

**Figure 7.** PDF and CDF of different statistical distributions (a-e, g-h). (f) shows the mixture model of the Log-Normal, Rayleigh, and Alpha-Stable distributions, which is used for the MSTA-CRF classifier and shows a good approximation of the original SAR histograms and the CDFs shown in (g-h).

### 4.4.1 Ice-water classification and validation

The example image shown in Figure 4 acquired on August 25, 2015 was used to evaluate the performance of the proposed MSTA-CRF algorithm. To validate the stability of the training dataset, the training samples were selected randomly from the dataset for 10 times, and the OA and the STD are reported in Table 6. We implemented the general CRF classifier, statistical distribution based CRF as well as the SVM classifier for comparison. The general CRF only contains the unary and pairwise potential (see equation 5 in section 3), the statistical distribution based CRF adds the statistical distribution potential to the general CRF classifier. The SVM classifier transforms the data into a higher dimensional space and implements the "one-against-one" technique by separating the nonlinear data using a RBF(Radial Basis Function) kernel (Zajhvatkina et al., 2017). A features ensemble including the GLCM (gray level co-occurrence matrix) is input into the SVM classifier. Then ice-water classification is calculated pixel by pixel. In the experiment. The GLCM features include mean value, standard deviation, energy, contrast, homogeneity, correlation and entropy. For the GLCM the backscatter values are discretized to 32 gray levels with the window size of $16 \times 16$ pixel. The separated distance is 8.

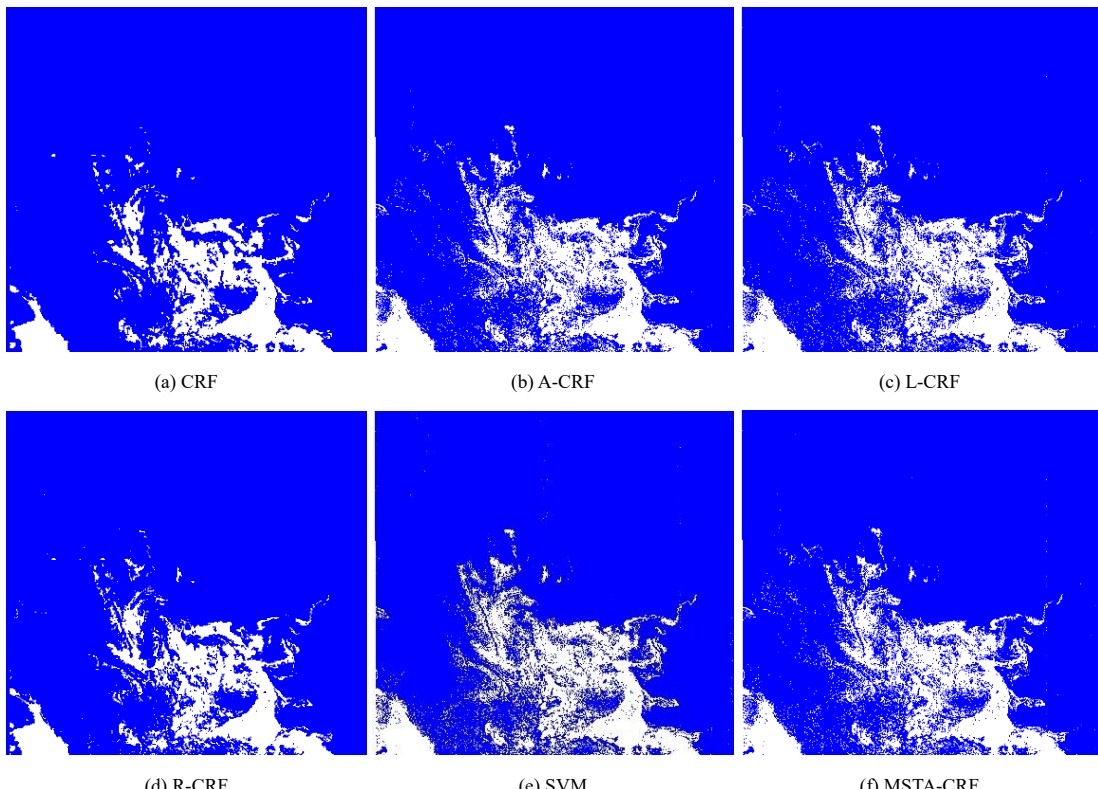

| (a) CRF | (b) A-CRF | (c) L-CRF |
| (d) R-CRF | (e) SVM | (f) MSTA-CRF |

**Figure 8.** Classification results of the data captured on August 25, 2015 using the different statistical model distributions shown in Figure 7 in the Connected Random Field (CRF) framework. A: Alpha-Stable L: Log-Normal, R: Rayleigh.

As shown in Table 7, the MSTA-CRF achieves the highest classification accuracy of 92% among these algorithms. Compared with the general CRF classifier, all statistical distribution based CRF classifier obtains higher classification results, which indicates that the statistical characteristics of SAR images are useful for ice-water classifications. For the different statistical distribution based CRF classifier, A-CRF has the highest OA. It is hard to say if the features based



SVM is better than the single statistical distribution based CRF classifiers since their results are similar. But the MSTA-CRF classifier utilize the mixture model as well as the fully connected semantic relationship and thus outperforms the SVM classifier in all four classification quality categories (Table 6). Moreover, the variation of the OA and STD for the randomly selection training samples is significantly lower than for the other algorithms.

**Table 7.** Overall classification accuracy (OA) and standard deviation (STD) of different algorithms, A: Alpha-Stable L: Log-Normal, R: Rayleigh.

| Algorithm | OA/% | STD/% | OW error/% | Ice error/% |
| --- | --- | --- | --- | --- |
| CRF | 84.73 | 6.22 | 5.96 | 7.31 |
| A-CRF | 88.02 | 5.73 | 3.25 | 7.03 |
| L-CRF | 85.47 | 6.44 | 6.29 | 8.22 |
| R-CRF | 87.21 | 9.03 | 5.75 | 7.04 |
| MSTA-CRF | 92.23 | 4.68 | 3.56 | 5.21 |
| SVM | 87.64 | 9.22 | 5.73 | 6.63 |

### 4.4.2 Ice-water classification comparison with ice charts

To validate the MSTA-CRF results we compare them to MET Norway ice charts, which is shown in Figure 9. For our purpose the ice chart classes "open drift ice", "close drift ice", "very close drift ice" and "fast ice" are defined as ice class.
And "open water" , "ice free" and "very open drift ice" are defined as water class. In Figure 9(c), it is clear to see that MSTA-CRF result shows a good agreement with the MET Norway ice chart in most cases. The main differences are located in boundary areas, most of the very open drift ice are misclassified as ice as a result of the difference in methodology of the ice chart. Larger areas with very open ice drift are naturally not recognized as ice areas by our higher resolution MSTA-CRF algorithm. Moreover, use of different satellite input data source may also lead to the discrepancies between our
classification results and the MET ice charts.

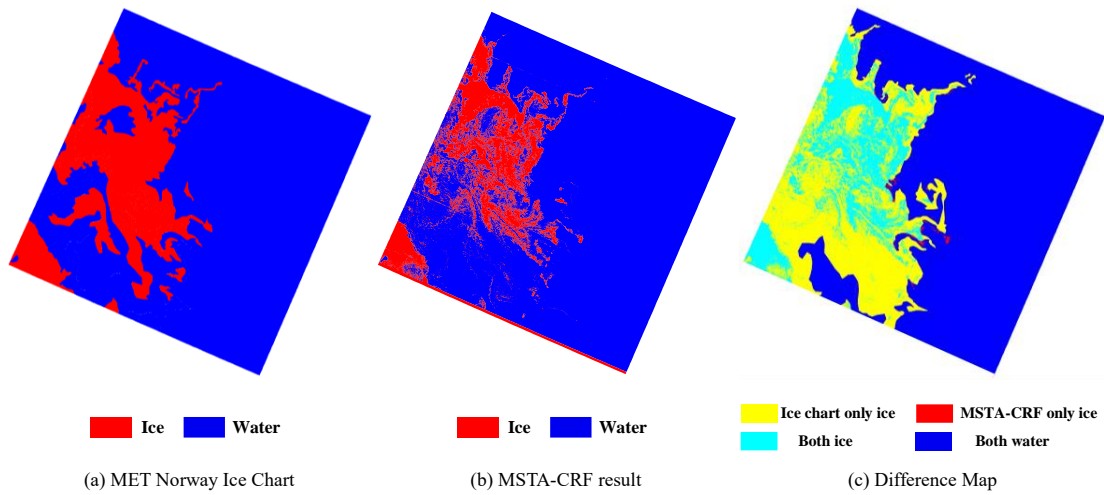

(a) MET Norway Ice Chart  (b) MSTA-CRF result  (c) Difference Map

**Figure 9.** Sea ice classification Validation of MSTA-CRF algorithm with MET Norway ice charts on August 25, 2015. Ice chart only ice means ice type on MET Norway ice chart but water on MSTA-CRF, MSTA-CRF only ice means water on MET Norway ice chart but ice on MSTA-CRF, both ice and both water means MET Norway ice chart and MSTA-CRF have the same category of ice and water respectively.

### 4.4.3 Calculation of sea ice concentration and comparison to ice charts



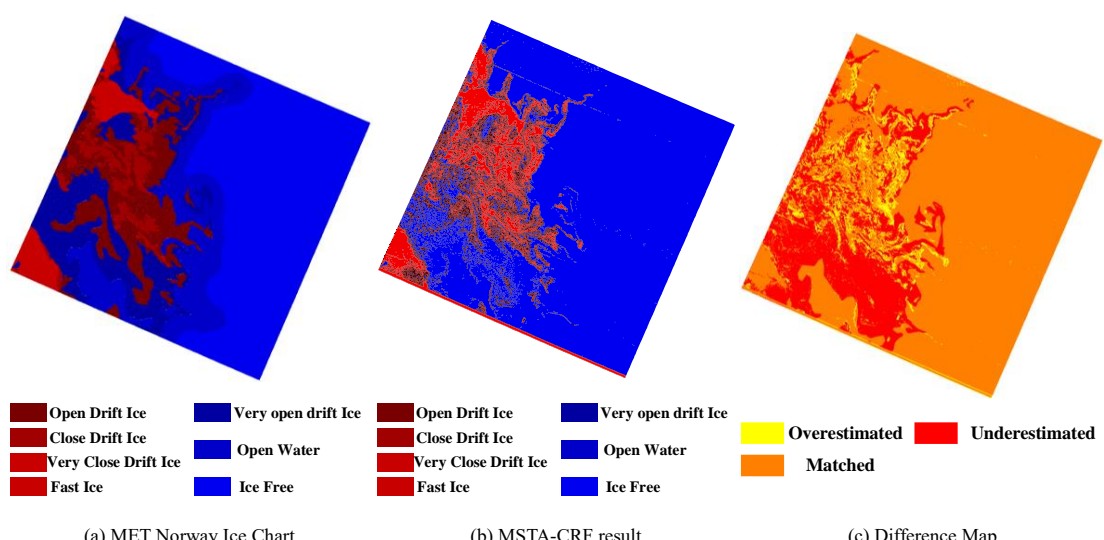

| | | | |
|---|---|---|---|
| ■ Open Drift Ice | ■ Very open drift Ice | ■ Open Drift Ice | ■ Very open drift Ice |
| ■ Close Drift Ice | ■ Open Water | ■ Close Drift Ice | ■ Open Water |
| ■ Very Close Drift Ice | ■ Ice Free | ■ Very Close Drift Ice | ■ Ice Free |
| ■ Fast Ice | | ■ Fast Ice | |

| | |
|---|---|
| ■ Overestimated | ■ Underestimated |
| ■ Matched | |

(a) MET Norway Ice Chart         (b) MSTA-CRF result         (c) Difference Map

**Figure 10.** Sea Ice Concentration Validation of MSTA-CRF algorithm with MET Norway ice charts on August 25, 2015. Overestimated means the SIC on MSTA-CRF is higher than MET Norway ice chart, underestimated means the SIC on MSTA-CRF is lower than that on MET Norway ice chart, and the matched means the SIC on these two products are the same.

After the sub-superpixel based ice-water classification, which provides an ice or water class label for every 40x40 m pixel, we calculate the sea ice concentration (SIC) on a 1 km² grid. The percentage of ice pixels within a 25x25 cell provides the SIC with 1 km spatial resolution based on the MSTA-CRF model. The classification is based on sub-superpixel (on average 24 pixels), i.e., not every pixel is independent. This leads to a discretization of the SIC for the 1 km grid cells of about 2% to 4% (mind that many of the sub-superpixels are cut in smaller areas by the 1-km gridding).

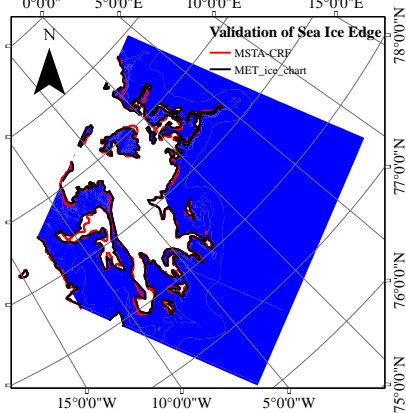

**Figure 11.** Comparison of the ice edge between the MSTA-CRF algorithm and the MET Norway ice chart. The ice edge defined in the MSTA-CRF is the boundaries where SIC lower than 15%, and the sea ice edge for MET Norway ice chart is the boundary of very open drift ice (1-4 of SIC in tenth).

The validation of MSTA-CRF sea ice concentration with ice chart on August 25, 2015 is shown in Figure 10. Similar to the ice-water classification comparison in Figure 9, the ice concentration errors are mainly located in boundary areas, and the overestimated and underestimated areas are mixed with each other Due to the lower spatial resolution of the ice



chart, some detailed information are lost. Besides, sea ice drift, which can be pronounced during melting season, may also lead to the difference between classification results and ice chart because their acquisition may be different.

The ice edge difference between the MSTA-CRF SIC result and the MET Norway ice chart are shown in Figure 11. The MSTA-CRF ice edge is defined as the 15% SIC isocline, and the MET ice edge indicates the boundary line between the "ice" and "water" category, which is defined in Table 2. It is clear to see that these two edges are matching very well in most cases. Only near [81.5N, 5E] and [79.5N, 12.5W] a slight difference can be seen. It also demonstrates that the proposed method is effective for sea ice and water classification during melting seasons. The detailed daily IIEE (Integrated Ice Edge Error) (Goessling, et. al., 2016) analysis will be discussed in Section 5.3.

**5. Results and discussion**

**5.1. Monthly Average Classification results**

We now apply the trained MSTA-CRF classifier to all Sentinel-1 SAR in our Fram Strait target region (Figure 1) for the six years 2015 to 2020 and calculate the accuracy by comparison to the daily MET Norway ice charts (Section 2.2.b). The monthly combined sea ice and water classification result in Figure 12 shows that the average accuracy for MAST-CRF algorithm is larger than 85%, and the STD is less than 10%, among which the OA in 2018 and 2020 is the highest of over 90%. We can also find that from the months June to September the accuracy in August always is the highest during the six years besides in 2015. Compared with the "OW" error, the "Ice" error is larger. The main misclassified area originates from melting water on fast ice, which occasionally is misclassified as water. Since we defined the very open drift ice as OW category, it may also lead to some misclassification.

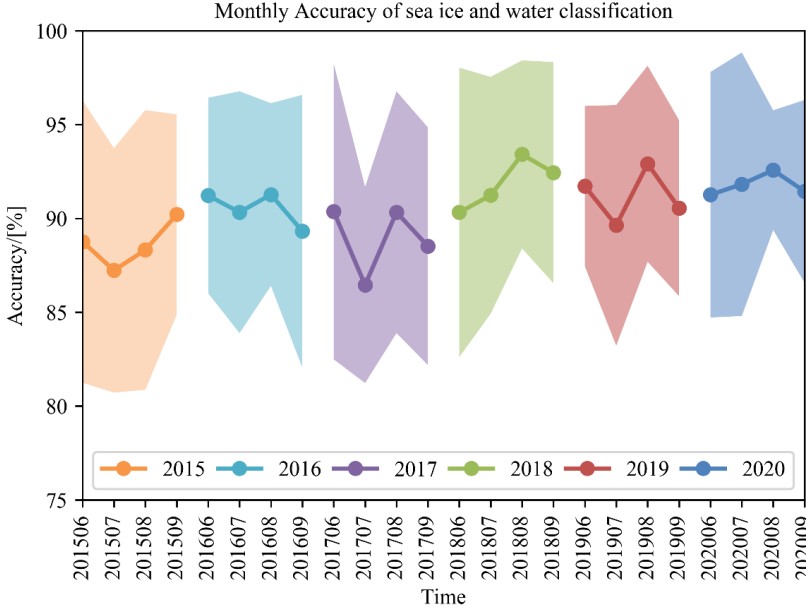

**Figure 12.** Monthly averaged OA (combined for "ice" and "water" class) and standard deviation of MSTA-CRF classification of S1 SAR scenes in the Fram Strait region validated using MET Norway ice charts. The value for each dot means the OA, and the corresponding range is the STD.

**5.2 Comparison of ASI sea ice concentration and MSTA CRF derived sea ice concentration**





Besides the MET Norway ice charts, the ASI SIC product from AMSR2 on a 3.125 km grid is also used to validate the performance of MSTA-CRF algorithm. For the comparison, the MSTA-CRF SIC is calculated on a 1 km grid from the ice-water classification (see section 4.4.3). Figure 13 gives two example results of SIC in the same area in 2020. One is acquired on June 1 and the other is on September 29, and these two images are for the same orbits, i.e., show the same

area. The MSTA-CRF sea ice concentration contains much more detailed information of sea ice conditions with the spatial resolution of 1 km. Especially for the lower SIC areas of leads and the marginal ice zone (MIZ), the MSTA-CRF SIC can detected more ice fragments. Moreover, MSTA-CRF SIC can also identify fragmented ice areas with many leads and lower SIC (central area in Figure 13e), which is some distance away from the sea ice edge. Overall, the SIC of the two products show very good agreement. This is confirmed in the scatter plots in Figure 13c and 13e, which show that the two SIC

results match well near 0% and 100%, and the difference of these two SIC products is less than 10% on average. Most of these areas are in the marginal ice zone or sea ice leads, where the MSTA-CRF SAR SIC has the advantage of a higher spatial resolution and can reproduce a higher SIC variability. There is also a tendency that the difference of classification result is caused by very open drift ice and open drift ice like for the MET Norway ice charts (in Figure 9 and 11). In conclusion, the MSTA-CRF using high resolution Sentinel-1 SAR data can provide much more detailed information of sea

ice conditions, while still reproducing similar SIC values as the ASI AMSR2 SIC.

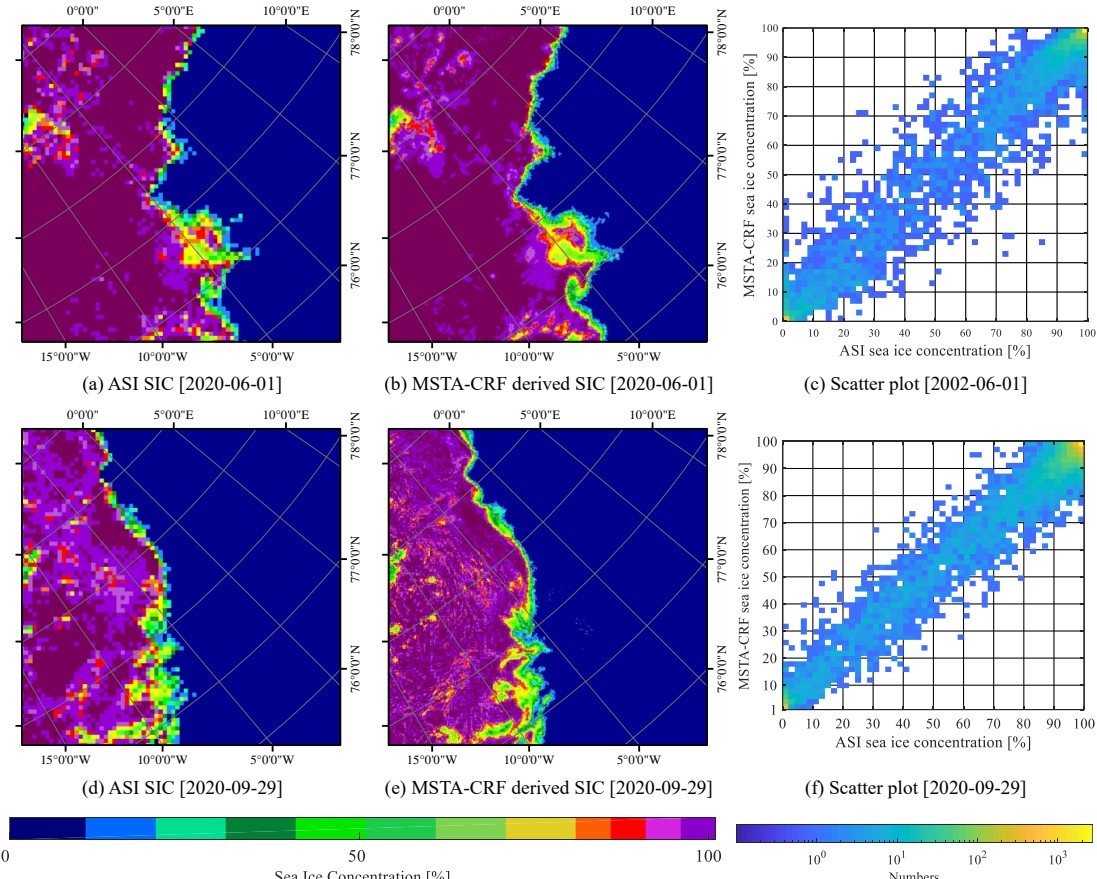

(a) ASI SIC [2020-06-01]      (b) MSTA-CRF derived SIC [2020-06-01]      (c) Scatter plot [2002-06-01]

(d) ASI SIC [2020-09-29]      (e) MSTA-CRF derived SIC [2020-09-29]      (f) Scatter plot [2020-09-29]





**Figure 13.** Comparison of ASI sea ice concentration and MSTA CRF derived sea ice concentration. The ASI SIC in (a) and (f) are 3.125km resolution, MSTA-CRF SIC in (b) and (e) are 1km resolution and the density of scatter map is in 3.125km by down sampling the MSTA-CRF SIC into 3.125km.

## 5.3 Time series of sea ice area during melting seasons and its variability

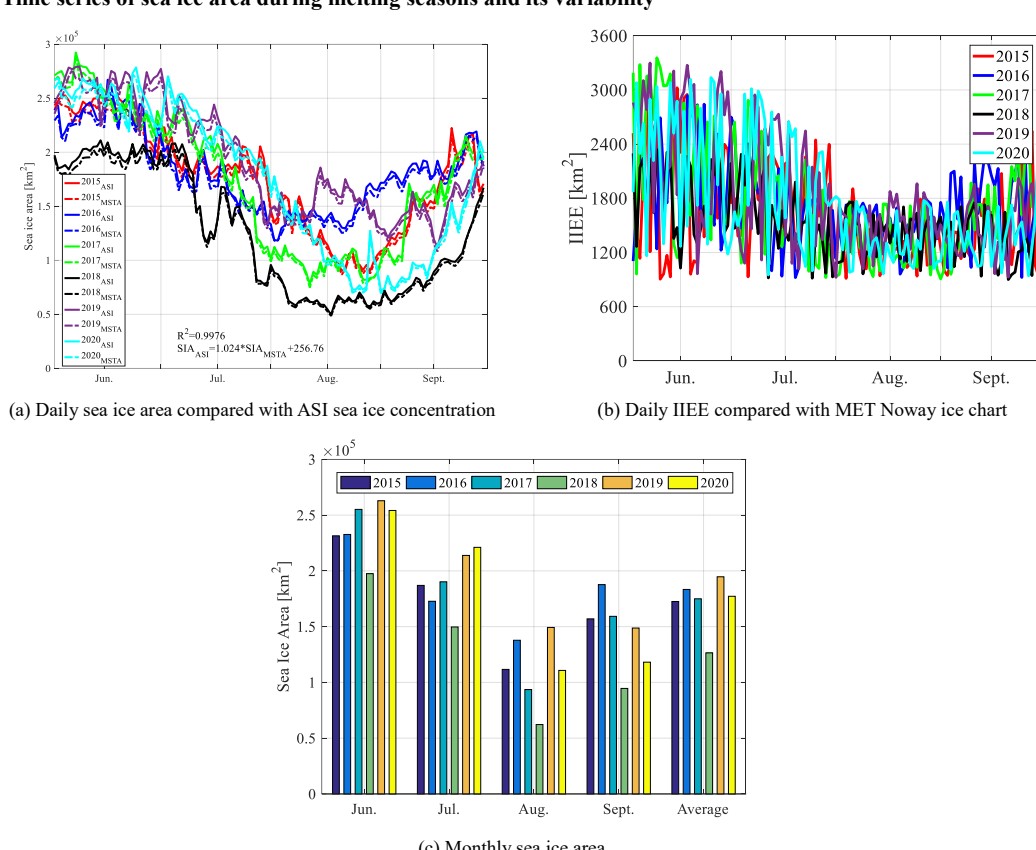

(a) Daily sea ice area compared with ASI sea ice concentration

(b) Daily IIEE compared with MET Noway ice chart

(c) Monthly sea ice area

**Figure 14.** Sea ice area during melting seasons from 2015-2020 in the Fram Strait region ([75N 83N], [-15W 15E]).

Here we discuss the daily MSTA-CRF SAR SIC time series for the Fram Strait region during summer for the six years 2015 to 2020 and compare it with the ASI AMSR2 SIC (Figure 14). In September 2020, the Arctic has experienced the lowest sea ice extents since the record minimum year of 2012 (https://nsidc.org/arcticseaicenews/charctic-interactive-sea-ice-graph/). For the Fram Strait region ([75N 83N], [-15W 15E]; Figure 1), however, the temporal development and seasonal cycle are different. The time series of daily sea ice area in Figure 14a indicate that the sea ice minimum for this region appeared in late August, 2018. During the four months June to September the sea ice area in Fram Strait usually is highest in late June and then decreases until its minimum in mid-August and then increase again from late August to September. Compared with the sea ice area from the ASI SIC product, the MSTA-CRF result is lower in many cases from June to mid July. While during the sea ice decrease period from July to early September, sea ice area calculated from MSTA-CRF is higher than ASI, the main difference is that thin ice may be misidentified as lower concentration using ASI method as well as the lower resolution of the dataset. The two results also shows a significant agreement with the $R^2$ of 0.9976. For the monthly sea ice area in 2018, the sea ice area in August only reaches the half of 2016. Figure 14b indicates



that the IIEE compared with MET Norway is can be as large as over 3000km² in the beginning of June, and then it decrease until the late August. After that, it stars to increase again. The averaged IIEE only 3300km, which is below 2% of the total sea ice area in the Fram Strait. The monthly average result in Figure 14c shows that sea ice area did not change so much in 2015-2017 and 2019-2020, while it has a significant decrease in 2018, it has decreased 28.75% compared with sea ice
area in 2019. The main reason for the sea ice minimal in 2018 may be due to the sea ice export from Fram Strait to lower latitude area which is caused by sea ice drifting.

## 6.   Summary and Conclusions

We have developed an algorithm for ice-water classification and sea ice concentration (SIC) retrieval using Sentinel-
1 Extra Wide swath (EW) mode data acquired over the Fram Strait region during melting seasons (June – September). Especially, during this time of the year sea ice surface melt and wind-roughened surface conditions make ice-water classification challenging. The classification scheme utilizes the SAR backscatter after corrections, which includes incidence angle normalization and thermal noise reduction in a MSTA-CRF (mixture statistical distribution based conditional random fields) framework. The mixture statistical distribution (Alpha-Stable, Log-Normal and Rayleigh) is
used to construct an additional probability potential for the classification in addition to the SVM potential and the fully connected pairwise potential. The mixture of statistical distributions can serve as a proxy for turbulent ocean wave conditions or calm water condition and to characterize ice and water. For each sub-superpixel, the proposed MSTA-CRF constructs a fully connected network to eliminate the uncertainty of small fragments of ice caused ice break-up or by the melting water on ice-covered areas during melting seasons. The proposed algorithm has been tested on Sentinel-1 dual
polarization SAR scenes for the Fram Strait region. The comparison with MET Norway ice charts shows that the proposed algorithm reaches an overall classification accuracy (OA) of about 90% with STD less than 10%.

To construct a reliable training dataset, 9760 patches from 488 S1 images in 2015-2018 are used for training, we first select twenty samples (ten for sea ice and ten for open water) with the size of 64×64 pixels to construct the training dataset. The proposed algorithm can distinguish between small open water leads and sea ice and very well preserves the ice edge
(averaged integrated ice edge error of below 2% of total ice area) by selecting 1000 patches from the training dataset. Validation of the ice-water classification result was conducted by comparing with ice charts. The results indicate that the main error is open drift ice misclassified as very open drift ice (defined as "water" in this paper). This may have caused some of the discrepancies between our classification results and the MET Norway ice charts.

The acquisition scheme of the Sentinel-1a (launch 04/2014) and b (launch 04/2016) constellation allows very frequent
coverage of the Fram Strait region. We have classified the sea ice area from Sentinel-1 almost every day for the six years 2015 to 2020 for the four summer months June to September. The sea ice area in Fram Strait shows a clear seasonal cycle in all years with the sea ice area minimum in August about one month earlier than the Arctic-wide sea ice minimum. Out of the four years the year 2018 clearly shows the lowest sea ice area during summer with an especially pronounced decrease from late August to September in 2018. In comparison to ASI AMSR2 microwave radiometer SIC our S1 SIC shows a
very similar seasonal cycle and close agreement in SIC values. However, the 1-km SAR SIC provides much more details, e.g. for leads and especially in the MIZ and for low ice concentration regions along the ice edge. We show that our MSTA-CRF classifier can provide very stable and accurate classification results throughout the summer. It can be extended towards winter season in future by further training. During summer no such high resolution SIC dataset exists. During



winter in a future study the SAR SIC can be compared with SIC from infrared imagers (e.g. Ludwig et al., 2020), which provide a similar spatial resolution but are not available during summer. To extend the MSTA-CRF S1 classification also to areas that are not as well covered by S1 acquisitions as the Fram Strait region, in future, they can be combined with passive microwave SIC datasets in an active-passive microwave fusion framework.

*Data availability.* All processed data can be obtained by contacting the first author. The SAR data used in this paper are downloaded from the European Space Agency website (https://scihub.esa.intdhus/). The Norwegian Ice Service at the Norwegian Meteorological Institute (MET Norway) provided daily sea ice charts for validation (delivered by Nick Hughes). The daily averaged wind speed, wave height and ice surface temperature are downloaded from ECMWF website.

*Author contributions.* The concept of the study was conceived by YZ and TZ. The SAR processing was done and reported jointly by YZ and TZ. CM, GS, and MH commented the SAR processing and methodology. NH provided the MET Norway ice charts and commented on the experiment part. YZ and GS wrote the conclusions section. All authors commented on the results. All authors contributed to the editing of the text and agreed to the submission.

*Competing interests.* The authors declare that they have no competing interest.

*Acknowledgements.* This work was supported by the national Key Research and Development Program of China under Grant 2017YFA0603104; the National Natural Science Foundation of China (No. 41801266, 41901275 and 41531069; the China Postdoctoral Science Foundation, No. 2017M612512 and LIESMARS special research funding. GS, CM and MH acknowledge supported by the Priority Program 1158 "Antarctic Research with Comparable Investigations in Arctic Sea Ice Areas" through the SITAnt project (project number 365778379), the Transregional Collaborative Research Center TRR 172 "ArctiC Amplification: Climate Relevant Atmospheric and SurfaCe Processes, and Feedback Mechanisms (AC)3" (268020496), and the MOSAiCmicrowaveRS project (420499875) all funded by the Deutsche Forschungsgemeinschaft (DFG).

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
