# Peer review of "Sea ice and water classification on dual-polarized Sentinel-1 imagery during melting season"

_The Cryosphere, 2021_

## Author Comment (AC1)

Dear authors of the manuscript tc-2021-85,

In the manuscript a widely studied topic has been studied. The method is computationally quite heavy but the results in classification are good. The manuscript is quite thorough, the data set and the evaluation are quite comprehensive. There are still some aspects which need to be taken into account before publishing this manuscript. In the following are my comments.

We are grateful to the reviewer for the constructive comments on our manuscript (tc-2021-85) entitled "Sea ice and water classification on dual-polarized Sentinel-1 imagery during melting season". We have addressed all the comments. Our point-by-point responses are attached below in blue, while the original Reviewers' comments are in black.

Thank you again for valuable comments on our manuscript.

Sincerely,

Test and training data set: It is not very clear how the data has been divided into independent training and test data sets. Evaluation should be performed using a test data set which is independent of the training data set, i.e. the training data set must be excluded from the test data set. Now division into these two independent data sets is not very clear to me. Please, in detail describe the division to independent training and test (evaluation) data sets to confirm the reader that they are independent.

Response: For MSTA-CRF model training, one Sentinel-1 SAR image on each day from June to Sept in 2015-2018 was selected. We first randomly select 10 samples of each category (sea ice and water) from the selected SAR images to construct the training and testing data set (9760 samples in total). During the training procedure, we randomly selected 100 samples for each category from the training and test dataset to train the model, and the rest of the samples is then used to verify its accuracy. If the accuracy is lower than 99%, 50 samples are added for each class to update the model, and these added training samples are removed from the test samples until the final classification accuracy on the test data is better than 99 %. We have repeated the training procedure ten times and found that when the number of training samples reaches 1000 the accuracy is over 99%. We have finally selected 1000 samples for training model (500 for each category), which accounted for 10.25% of the entire training and testing data set. The following table describes the procedure of MSTA-CRF model training. The table and the corresponding flowchart below will be included in the next version of the manuscript.

| Step 1 | SAR image selection

One SAR image on each day from June to Sept in 2015-2018 is randomly selected to construct the training data set, finally we get 488 images. |
|--------|----------------------------------------------------------------------------------------------------------------------------------------------|
| Step 2 | Training and testing data set construction: |

| | |
|---|---|
| | 10 patches (samples) for each category (ice and water) with the size of 64*64 pixels are randomly selected from the 488 SAR image using MET Norway ice charts, then we get 9760 patches for constructing the training and testing data set. |
| Step 3 | MSTA-CRF training:

100 patches for each category are selected for training the MSTA-CRF model, and the rest are used as testing samples to decide by the overall accuracy whether the training will be repeated. |
| Step 4 | Testing:

If the overall accuracy of the testing samples is larger than 99%, then we get the final MSTA-CRF model, otherwise 100 patches (50 for each category) will be added to retrain the MSTA-CRF model, and the newly selected 100 patches will be removed from the testing samples. |
| Step 5 | SAR image classification:

Repeat step 3 until we train a satisfied model, and the newly trained model will be used for sea ice and water classification on all the SAR images. |

We also give the flowchart of the training procedure in the following figure.

[Figure]

Introduction:
Also sea ice concentration (SIC) estimates can be and are derived based on the proposed SI/OW classification scheme. I recommend to include missing references to SAR-based SIC estimation, there are many papers on this published during the recent years, e.g.:

Wang, L., K. A. Scott, L. Xu, D. A. Clausi, Sea ice concentration estimation during melt from dual-pol SAR scenes using deep convolutional neural networks: A case study, IEEE Trans. Geosci. Remote Sens., vol. 54, no. 8, pp. 4524–4533, 2016.

Wang, Scott, Clausi, Sea Ice Concentration Estimation during Freeze-Up from SAR Imagery Using a Convolutional Neural Network Remote Sens. 2017, 9(5), 408; https://doi.org/10.3390/rs9050408

W. Aldenhoff, A. Berg and L. E. B. Eriksson, "Sea ice concentration estimation from Sentinel-1 Synthetic Aperture Radar images over the Fram Strait," 2016 IEEE International Geoscience and Remote Sensing Symposium (IGARSS), 2016, pp. 7675-7677, doi: 10.1109/IGARSS.2016.7731001.

Karvonen, Evaluation of the operational SAR based Baltic Sea ice concentration products, Advances in Space Research 56(1), 2015, DOI: 10.1016/j.asr.2015.03.039

And some references combining microwave radiometer and SAR for SIC estimation:

Karvonen, J., Baltic Sea Ice Concentration Estimation Using SENTINEL-1 SAR and AMSR2 Microwave Radiometer Data, IEEE Transactions on Geoscience and Remote Sensing (Volume: 55, Issue: 5, May 2017), pp. 2871-2883, 2017, DOI: 10.1109/TGRS.2017.2655567.

Malmgren-Hansen, D., Pedersen, L. T., Nielsen, A. A., Brandt Kreiner, M., Saldo, R., Skriver, H., Lavelle, J., Buus-Hinkler, J., Harnvig, K., A Convolutional Neural Network Architecture for Sentinel-1 and AMSR2 Data Fusion. IEEE Transactions on Geoscience and Remote Sensing, v. 59, n. 3, pp. 1890-1902. 2021, https://doi.org/10.1109/TGRS.2020.3004539

Especially convolutional neural networks in sea ice classification and parameter estimation have gained popularity during the recent years. These methods are computationally heavy but software for their parallel efficient execution on graphics adapters exist.

Response: Thanks for your comments. We will include some summary of the papers about sea ice concentration retrieval based on microwave data fusion, convolutional neural network based sea ice classification, as well as the above mentioned papers in the revised version.

P4 2.1 Research area:

The sentence "To consider the spatial contextual information and preserve the spatial details of each pixel in SAR imagery, the energy function based maximum a posteriori (MAP) estimation in MSTA-CRF framework is proposed for operational ice water classification during melting seasons in Fram Strait." does not belong to this subsection, it could be in introduction or methodology section rather. Just start the section by "This study was performed in the area of Fram Strait during the melting season." or something similar.

Response: We agree with the comments, and we will rewrite the sentence in the manuscript.

P4 L21: "Figure 1 shows an overview of the research area and some satellite scenes used in this manuscript." and Figure 1 / Figure 1 caption.

Why just some scenes are shown? Could the figure for example show the total amount images at each location of the study area (by using some color coding), it would be much more informative.

[Figure]

Response: We provide a new figure 1 in the revised manuscript. The different colors of the rectangles indicate the SAR images acquired in different years, and the number in the rectangle on the top right of the figure means the number of SAR images used for sea ice classification in the corresponding year.

P5 Sentinel-1 SAR Data:

L6: "... data during melting seasons from 2015 to 2020 are used.". Please, be more specific, give the periods. Is the melting period the same every winter? E.g. some kind of temperature statistics from nearby weather stations to confirm that the data represents melting period every winter would be useful here.

Response: It is not our objective to define the melting season, our purpose is to propose an algorithm for sea ice classification in melting seasons, and it can also be used for sea ice classification in other seasons. To analyze the melting condition in Fram Strait, we have downloaded the hourly averaged ERA5 2-meter temperature (spatial resolution of 0.25°) from the ECMWF website. We illustrate the daily averaged temperature in Fram Strait in Figure 2. It is clear to see that the temperature starts to increase from the beginning of June, and reaches top in the beginning of August, then it starts to decrease and finally drops below 268K at the end of September. Except for the year 2015, when the surface temperature falls below 273K at the beginning of September, in 2016-2020 the surface temperature falls below 273K in the middle of September. A surface temperature above 273K means that the sea ice is still in the melting condition. As a result, we have selected the Sentinel-1 SAR data from June to September each year and defined this time span as the melting period in Fram Strait.

[Figure]

P6 Methodology:

Figure 2. If I have understood correctly SPAN image is used as an input? Now in the figure there is an arrow from the leftmost SAR processing block to the MSTA-CRF block and it looks like the uppermost row SAR data were input to the MSTA-CRF, possibly the arrow could be started from the lower part of the block as is the second arrow. Assuming I have understood this correctly.

Response: You are right. We have revised Figure 2.

[Figure]

P5 L5: SPAN of the HH and HV channels (sqrt(HH^2 + HV^2)). Later SPAN is defined as square root of sigma0_HH^2 + sigma0_HV^2. Possibly the square root could be dropped from here and just say that SPAN represent the joint total power of the two SAR channels and leave the more precise definition later.

Response: You are right, it is sqrt(HH^2 + HV^2).

P9 L19: "using the MET Norway ice charts". Please, be more specific and describe exactly how the ice charts have been utilized.

Response: MET Norway ice charts provide manually classified sea ice categories daily. The reference (Zakhvatkina, 2017) also uses the MET Norway ice chart products for training and verification of sea ice classification. Therefore, we chose the MET Norway ice chart in the paper for sample selection and validation. In order to improve the accuracy of sample selection, we have also combined the visual inspection to improve the accuracy of sample selection. Besides, we have also analyzed in the paper that due to the difference between the SAR data acquisition time and the MET Norway ice chart acquisition time, the drift and freeze-thaw changes of the sea ice also affect the classification accuracy

P14 Fig. 6: Be more specific in Y-axis label, now there is just "normalized". "normalized" what? I guess "normalized parameter" would be better here. Fig. 6a is not very clear with so many curves in one figure. Would there be any alternatives to make a more clear image (or more than one image)?

Response: We have revised Figure 6(a), where in the top left we only give the normalized parameters of Weibull and Gamma (these two models are not used in MSTA-CRF modelling), and the remaining three models in the bottom left. The normalized parameter means that we have adjusted the value of the parameter to the range from 0 to 1.

[Figure]

(a) Parameter of single distribution models in Mixture model

(b) Weight coefficient of single distribution in Mixture model

P13 23: "CV (coefficient of variance)"? Do You mean "coefficient of variation"? At least for me coefficient of variance is an unknown concept. If You use it, please, define it.

Response: You are right. CV in the manuscript means coefficient of variation, and it is defined as the ratio of the standard deviation and the mean accuracy, as can be seen in the following reference, and it will be added in the next version of the manuscript.

Keller M R, Gifford C M, Winstead N S, et al. Active/Passive Multiple Polarization Sea Ice Detection During Initial Freeze-Up[J]. IEEE Transactions on Geoscience and Remote Sensing, 2020.

Minor:

P9 L13: "Figure 5 (e)", You probably mean Figure 4 (e)?

Response: You are right. It is Figure 4(e).

P9 L18 "for each category", as there are only two categories it would be better to say "for both categories".

Response: We have corrected it.

P10 L2: "each categories" -> "both categories"

Response: We have corrected it.

P10 L10: "...that when the training samples reaches..."? Do you mean "...that when the number of training samples reaches..."?

Response: We mean that when the number of training samples reaches 1000 (500 samples for each category) the overall accuracy is over 99%.

P10 L32: "...contains a great number of scatters of radiation..." -> "...contains a large number of scatterers of radiation...".

Response: We have corrected it.

P11 Table 3 caption: "...probability density function (pdf)..." -> "...probability density function (PDF)..." or rather even "...probability density function..." and give the acronym PDF in the text.

Response: We have corrected it by using probability density function (PDF).

P11 L12: "PDF" -> "probability density function (PDF)", PDF always with capital letters.

Response: We correct it by using probability density function (PDF).

P11 L22 and Eq. 8: Explain what is M (is it number of PDF's here?).

Response: M in the equation means the number of PDFs. In the manuscript, we finally use three different PDFs including Alpha-Stable distribution, Log-Normal distribution and Rayleigh distribution.

P12 L6: "...into several sub-superpixel using a random number..."? What does this mean? "...into several sub-superpixel using a random number of pixels..."?

Response: You are right, the superpixel will be segmented into several sub-superpixels using a more restrict mean-shift procedure, and the maximal number of sub-superpixel within a given superpixel is less than 50.

P12 Eq. 9: What are K and N in the equation?

Response: K is the number of kernels in the fully connected network and N is the number of pixels. We will add this information to the text.

P12 "...$y_i$ and $y_j$ are the SAR backscatter coefficients at a pair of subsuperpixel..."? Are these really SAR sigma0 values or SPAN values?

Response: it is sigma0.

P14 Fig. 6: Add x-axis labels ("model number" or something describing what is on the x-axis).

Response: We add it.

P15 L17-18: "PDF (probability density function)" This has already been opened on p. 11, so just write "PDF".

Response: we remove it.

There seem to be some sentences which are not very easy to understand. I am not a native English speaker and may not have noticed all of these sentences or possible grammar or typing errors. I recommend to let a native English speaker (Your co-author Nick Hughes) to check the sentences and language of the revised manuscript before submission.

Sincerely,

---

## Author Comment (AC3)

The authors detail a SAR image sea ice-water classification technique for use during melting conditions in the Fram Strait region. The input data are dual-polarization (HH + HV) Sentinel-1 EW mode scenes which are widely available over marine regions and open access. Their method for pre-processing of the HV channel of the Sentinel-1 SAR data seems to work very well, enabling its inclusion in the classifier. A good classification accuracy of ~90% is achieved, and the results are used to examine sea ice concentration evolution in the summer months over the 2015-2020 period. Since C-band SAR images are commonly used for ice mapping and charting, the results are potentially extendable to other missions as well. The potential to use a SAR based sea ice concentration algorithm during the summer months, and in a marginal ice zone, when/where passive microwave data is less reliable, is also noteworthy.

We are grateful to the reviewer for the constructive comments on our manuscript (tc-2021-85) entitled "Sea ice and water classification on dual-polarized Sentinel-1 imagery during melting season". We have addressed all the comments. Our point-by-point responses are attached below in blue, while the original Reviewers' comments are in black. Thank you again for the valuable comments on our manuscript. We will go through it and revise the certain parts of the manuscript following the reviewer's suggestions.

1. The paper is hard to follow, especially given that there is a lot of repetition in the text and figures, and some concepts and ; acronyms defined more than once. The input data to classification result is shown in Fig. 2 and Fig. 5; the SAR processing to remove noise is shown in Figs. 2, 3, and 4. Training sample selection is detailed in Sections 3.2. and 4.2. CRF and MSTA-CRF are defined on Page 3 then defined again on Page 6 (etc.). The selection of reference incidence of 23° doesn't need to be introduced on Page 8 then again on Page 13. The authors should describe their methodology in terms of input data, pre-processing, training, classification, and validation, and make it shorter in length. Everything on Page 17 and later could be included in Results and Discussion.

   Response: We have tried to rewrite it according to your suggestions.

2. It is unclear what input data is actually used. Fig. 1 shows some scene extents though it is difficult to tell whether they are arbitrarily chosen or what they are supposed to represent. Later in the paper there is mention of 488 images, or one image each day from June to September over the period of 2015-2018. Provide more detail on what Sentinel-1 data are used (without listing them).

   Response: We provide a new figure 1 in the revised manuscript. The different colors of the rectangles indicate the SAR images acquired in different years. Even this figure cannot show all the images used for classification, therefore, the

number of images used for each year is given on the top right of the figure.

3.  The images are described as pertaining to melting conditions. More justification for this should be provided since it is insufficient to assume that all images between June and September are in melting conditions at this latitude.

[Figure]

Response: You are right. Not all the area is melting in this latitude. We not only can solve the problem in the melting season, but also get good results in other situations.

We downloaded the hourly averaged ERA5 2-meter temperature from ECMWF website with spatial resolution of 0.25°. Then we illustrate the daily averaged temperature of Fram Strait in Figure 2. It is clear to see that the temperature starts to increase from the beginning of June, and reaches its top at the beginning of

August, then it starts to decrease and finally drops below 268K at the end of September. Except for the year 2015, when the surface temperature dropped below 273K at the beginning of September, in the years 2016-2020 the surface temperature drops below 273K in the middle of September. Surface temperatures above 273K mean that the sea ice is still in melting condition. As a result, we select the Sentinel-1 SAR data from June to September each year and define this time span as the melting period of Fram Strait.

4. If there is a Sentinel-1 image from each day in the Fram Strait, images that correspond more closely to the MET Norway ice charts should be used for selection of training data. Otherwise there is more chance for ice drift and changing ice/water conditions to introduce error into the training sample selection.

Response: This is correct. Training procedure will also make some influence on the classification result since the selection of training samples depends on the ice chart. But we do not have the detailed information of acquisition times of the datasets used for the ice charts. The maximum time difference would be 24 hours but as Sentinel-1 data is also used for the ice charts the actual time difference in many cases will be smaller.

The MET Norway ice charts are currently the only sea ice product that can be obtained with a temporal resolution of one day. The reference (Zakhvatkina, 2017) also uses the MET Norway ice chart products for training and verification of sea ice classification. Therefore, we chose the MET Norway ice charts in the paper for sample selection and validation. In order to improve the accuracy of sample selection, we have also combined the visual inspection to improve the accuracy of sample selection. Besides, we have also analyzed in the paper that due to the difference between the SAR data acquisition time and the MET Norway ice chart acquisition time, the drift and freeze-thaw changes of the sea ice also affect the classification accuracy

5. Does the inclusion of GLCM features in the SVM classifier improve its performance when compared to using just HH + HV data? The inclusion of GLCM features is described but it is unclear why, and on what basis the GLCM parameters, the kernel size, quantization level, and displacement were chosen

Response: The SVM classifier is used for two very different purposes in the manuscript. 1) It is part of our model. Here, we do not use the GLCM, only the HH+HV. We agree that the accuracy could potentially be improved by including GLCM but we did not test that. 2) The SVM is used as the comparison method (Zakhvatkina et al., 2017) for the classification accuracy, and it will be used as the comparison method in this manuscript. This SVM is used and implemented in the same way as described in Zakhvatkina et al. (2017)

In the paper about the SVM algorithm, Zakhvatkina et al.(2017) use Radarsat-2 data to achieve a good detection effect (~90%), but the classification accuracy is poor (~75%) in summer. In our manuscript, the SVM classifier is used to demonstrate the performance of our approach and we use the same configuration which was used by Zakhvatkina et al., where the GLCM features include mean value, standard deviation, energy, contrast, homogeneity, correlation and entropy

with the 32 gray levels. The window size for GLCM calculation is 16*16 pixels, and the separated displacement is 8.

6. The main misclassification error, on a class-by-class basis, is given to be caused by the presence of melting water on fast ice, leading to misclassification of ice as open water. However it is unclear how this was determined. If it is assumed, then the authors should provide some justification for it (e.g. article reference).

Response: Sorry for the misunderstanding. Fast ice is defined in the MET Norway ice chart, but in figure 10, there is no fast ice, and we will remove it. The brightest is the "Very Close Drift Ice", and there is no fast ice category in Figure 10.

The difference map in Figure 10(c) is mainly caused by the underestimation of the SIC compared with the MET Norway ice chart. As shown in Figure 10(a) and (b), the reason for the underestimation is that open drift ice (sea ice) is misclassified as very open drift ice (sea water). The averaged sea ice concentration of open drift ice in Table 2 is about 5.5 (calculated from 0 to 10), which means open drift ice also contains open water, especially the influence of sea ice surface melt water during the melting period, resulting in this underestimation.

7. Consistency in terminology is needed, e.g. "backscatter", "backscatters", "backscattering", and "backscatter coefficient"; "incidence angle" and "incident angle"; "RS-2" and "RS2" etc.

Response: We have revised it.

Specific comments:

(Page = P, Line = L)

P1L23: "backscatters" should be "backscatter"

Response: corrected

P2L21: delete "value"

Response: corrected.

P2L23: should be "MAp-Guided"

Response: corrected.

P2L30: data "are" (plural)

Response: corrected.

P2L32: Backscatter is also affected by waves that form, e.g. by capillary action, not just waves propagating into the area.

Response: We reformulate the sentence as follows:

"Backscattering characteristics are determined by sea ice surface roughness and its dielectric properties. As these are different for ice and water, backscatter can be fully explored to separate different sea ice types and water. However, the backscattering can also can be affected by ocean waves, mainly by small scale capillary waves for open water areas like leads but also swell propagating into the ice area, which can cause ambiguities. Therefore, it is not enough to only rely on backscattering intensity for identifying the different ice types and water."

P3L2: Delete "scatters of the"; also change "the mixture" to "a mixture" on the next line.

Response: corrected.

P3L7: "analysis" should be "classification"

Response: corrected.

P3L9: "Texture"

Response: corrected.

P3L14: Use of the term "usually" here creates ambiguity.

Response: corrected.

P3L31: "A statistical distribution …."

Response: corrected.

P4L15-18: There is a lot of detail given about the classification method here. The focus should be on Fram Strait.

Response: rewritten.

Fig. 1.: Make a better map with the image detail provided.

Response: corrected.

P5L6: "mode"

P5L15: Very Open Drift

is defined as SIC<1 here, whereas in Table 2 it is shown as 1-4.

Response: We have corrected this. Very Open Drift is defined as SIC in the range 1-4.

P7L12: "sub-swaths"

Response: corrected.

P7L18-19: Delete sentence that starts "Preprocessing methods …"

Response: deleted.

P8L9-10: Delete the description "with SPAN being defined…" etc. since the equation is given. The equation doesn't need to be in Fig. 4.

Response: deleted.

P10L3: Delete "e.g." and correct "otherwise"

Response: corrected.

P10L32: "noise is based on…"

Response: corrected.

P12L8: Sentence beginning "It may have" is hard to understand. Perhaps break it up.

Response: corrected.

P13L28: The table isn't really necessary since the analysis and its outcome is described well above it.

Response: deleted.

P15L31: "and Weibull distributions are not …"

Response: corrected.

P17L10: Delete "In the experiment"

Response: deleted.

P18L12: Provide some detail about the temporal offset between the classification result and the ice chart.

Response: You are right that the evaluation of the classification could be affected by the temporal offset between SAR image and ice chart, but we do not have the detailed information. The MET Norway ice chart is a daily averaged product, but it is only provided from Monday to Friday.

In melting seasons, the drift speed of sea ice is accelerated and the ice condition may change a lot in a few hours. Due to the different acquisition times of the SAR image and the data used for the MET Norway sea ice charts, this may lead to uncertainty of the validation results. Figure 12 illustrates the daily average SST

(Sea Surface Temperature), SWH (Sea Wave Height) and wind speed of the Fram Strait using ECMWF ERA Interim data. It is clear to see that the ocean conditions have changed a lot within six hours. For the SAR image acquired on 16:23:33, 25 August, 2015 (Imaging in one minute), the SST remains stable from 12:00 to 18:00, While the V wind component was lower in the whole southern part of the Fram Strait at 12:00, at 18:00 high wind speed was in the eastern part. We can clearly see that sea ice shows a trend of drifting from west to east. The direction of the U wind component has reversed the sign, and the wind component has rotated by 90 degrees clockwise from 12:00 to 18:00. As a result, the SWH was lower in the northeast at 12:00 and became higher at 18:00. Moreover, the SWH of zero means that the data are missing, since SST is the lowest (-2℃), it indicates that these areas are mostly covered by sea ice.

[Figure]

P20L13: "MSTA"

Response: corrected     .

P21L4: "from the same orbits"

Response: corrected     .

P22L3: delete "has"
Response: corrected     .

---

## Author Comment (AC5)

This manuscript introduces a new method for the ice mapping based on dual-polarized Sentinel-1 SAR data during summer season. The proposed method was developed from a conditional random field based on mixed statistical distribution. The results indicate the potential to derive reliable ice extent operationally. Unfortunately the author's use of English is very poor, and the meaning was ambiguous and confusing in most cases. Missing methodological details, incorrect use of models, and the large number of typo and formatting errors do not make an impression of a self-contained manuscript. The authors should not submit a manuscript which is not ready for submission. I would recommend a rejection of this paper, but I think that the author could have a chance of publishing the results of their study if they prepare the manuscript better next time.

We are grateful to the reviewer for the constructive comments on our manuscript (tc-2021-85) entitled "Sea ice and water classification on dual-polarized Sentinel-1 imagery during melting season". We have addressed all the comments. Our point-by-point responses are attached below in blue, while the original Reviewers' comments are in black. Thank you again for valuable comments on our manuscript. We will go through it and revise the parts of the manuscript following the reviewer's suggestions.

General Comments:

1. The description of the methodology is so poorly structured, which makes the logic of the research very confusing and hard to follow. One example is that the description of data preprocessing and training samples selection should not be introduced in the section of methodology. I am pretty sure a lot of efforts are still needed for improving the general structure of the paper.

Response: Thanks for your comments and suggestions, we will rewrite the manuscript to make it more understandable, but we think the training sample selection and preprocessing are also the key steps of the sea ice and water classification procedure, and so they actually fit into the methodology section.

However, we will add a clearer description of the training procedure, the definition of research area and the data used, you can see it in the response to reviewer 1 as well.

2. The training samples was selected from the MET Norway ice chart. The MET ice chart is a weekly product and it inevitably has a time lag with SAR data. The change of sea ice is fast in melting period. How do you make sure the samples you choose are correct?

Response: The MET Norway ice charts are a daily averaged product, but are only available from Monday to Friday, and it can also be used for daily sea ice classification. We select the samples using a combination of MET Norway charts and visual inspection. The MET Norway ice charts are currently the only sea ice product that can be obtained with a temporal resolution of one day. The reference (Zakhvatkina, 2017) also use the MET Norway ice chart product for training and verification of sea ice classification. Therefore, we chose the MET Norway ice charts in the paper for sample selection and validation. In order to improve the

accuracy of sample selection, we have also included  visual inspection. Besides, we have also analyzed in the paper that due to the difference between the SAR data acquisition time and the MET Norway ice chart acquisition time, the drift and freeze-thaw changes of the sea ice also affect the classification accuracy.

3. In the step of incidence angle correction, the authors used an incorrect sea ice scattering model. In equation (1), the backscattering of sea ice is described as the function of nadir backscattering and $\cos^n(\theta_i)$. When the radar echo is incident vertically, the scattering mechanism of sea ice is specular scattering which is completely different from the scattering mechanism of SAR. Therefore, the used approach is illogical and unphysical.

Response: You are right. The equation (1) is removed from the manuscript, we only use the measured backscatter value. Usually the backscatter depends on the incidence angle, and for Sentinel-1 SAR images with an incidence angle larger than 15°, the backscatter decreases with increasing incidence angle. The scattering mechanism of sea ice includes volume scattering (relevant for multi-year ice). We refer to a publication from TGRS.

W. Lang, P. Zhang, J. Wu, Y. Shen and X. Yang, "Incidence Angle Correction of SAR Sea Ice Data Based on Locally Linear Mapping," in IEEE Transactions on Geoscience and Remote Sensing, vol. 54, no. 6, pp. 3188-3199, June 2016, doi: 10.1109/TGRS.2015.2513159

4. The mean-shift method is critical to the proposed classification method. But the principle of mean-shift algorithm and the parameter setting for unsupervised segmentation should be introduced.

Response: The mean-shift method is a classical segmentation method in image processing. We did not introduce the details of the mean shift algorithm in this paper too much, but gave the mean-shift parameters setting. You can find detailed information on the mean-shift method in the reference, and it will be included in the next version. The principle of the mean-shift algorithm is to first define an offset value of the backscattering coefficient, and define the point where the difference between the backscatter coefficient and the current pixel point meets the offset value as the same clustering unit. In this paper, we give the parameter settings for mean-shift over-segmentation. For example, if the number of pixels within one superpixel is greater than 5000, and considering the heterogeneity of superpixels, In our manuscript, we define sub-superpixels of smaller units for each superpixel. We calculate about 100,000 sub-superpixels among them, and the average size of one sub-superpixel is 24 pixels.

Lang F, Yang J, Yan S, et al. Superpixel segmentation of polarimetric synthetic aperture radar (sar) images based on generalized mean shift[J]. Remote Sensing, 2018, 10(10): 1592.
Ming D, Ci T, Cai H, et al. Semivariogram-based spatial bandwidth selection for remote sensing image segmentation with mean-shift algorithm[J]. IEEE Geoscience and Remote Sensing Letters, 2012, 9(5): 813-817.

5. What I am most dissatisfied with is the use of distribution models. The distribution model of Gamma, Weibull and Alpha-stable is based on the statistical characteristics of pixels. However, the distribution model was for "sub-superpixel" (patches derived from unsupervised segmentation method) not for pixels. I don't think these distribution models could be adaptable to image patches.

Response: we are sorry for the confusion. We think the reviewer has misunderstood this part. We will improve the sentence in the manuscript. In fact,

the statistical distribution is fitted to the distribution of all the pixels within one sub-superpixel. Meanwhile, the statistical distribution is used as an input feature for super-pixels or sub-superpixels to calculate their potentials.

6. There are many SAR sea ice classification methods, taking these methods as baselines and comparing them with your method is necessary for validating the effect of your method. Moreover, the authors claimed that the advantage of proposed method is to identify sea ice in melting season. So you should give more examples to prove that the developed method can solve the problem of sea ice classification in summer.

Response: In this paper, the statistical distribution based CRF is proposed for sea ice and water classification. In order to verify the effectiveness of the algorithm proposed in this paper, the SVM algorithm (Zakhvatkina et al, 2017) is used as the comparison method in this manuscript. In their study, Zakhvatkina et al. (2017) use Radarsat-2 data to achieve a good detection effect (~90%), but the classification accuracy is poor (~75%) in summer. This paper focuses on the sea ice classification in summer, and the result shows that the method in this paper has better seasonal adaptability than the SVM method, and the classification accuracy in summer can reach about 90%. Compared with ASI sea ice concentration in summer (figure 13 and 14), our method provides satisfying results. To our knowledge, SAR images have so far been a focus in winter, now we are using SAR images to solve the problem of sea ice classification in summer.

7. According to the results of Table 4, the classification accuracy depends on used reference incidence angle. In equation (2), $\cos^n(\theta_{ref})$ is a constant value. I don't understand why the variation of constant value has an impact on classification accuracy.

Response: Thanks for your comments. Varing the reference angle will change the backscatter value, which will also have an influence on the MSTA-CRF training, and the classification result will also be different. Referring to table 4, the selection of the incidence angle has little effect on the classification accuracy. However we have to select one reference angle for classification.

In the incidence angle correction, we transform the backscattering coefficient of the entire image in the Sentinel-1 data to the reference incident angle $\theta_{ref}$, and it will be used as the input data for the sea ice classification experiment. In order to verify the optimal $\theta_{ref}$, we have designed the experiment of reference angle selection. In Table 4, the $\theta_{ref}$ is set in the range of 20-40 degrees, and the CV is used as the criterion. The result shows that the optimal reference angle is 23°. Therefore, all the classification experiments in this manuscript have corrected the backscatter of the original image to 23° in order to obtain the final classification result.

8. How to determine the parameters used in the proposed method (e.g. n and weight coefficients) is not clarified. Many details are not clear and need further explanation.

Response: We will try our best to describe this in more detail. The parameter estimation of the mixed statistical distribution adopts the EM (Expectation Maximization) method, which can be found in the paper by Tadjudin (2000). The general idea is to first calculate the distribution parameters of each statistical model, and then estimate the weight parameters of each distribution in the mixed distribution.

Tadjudin, S., and Landgrebe, D. A.: Robust parameter estimation for mixture model, IEEE T. Geosci. Remote, 38, 439–445, https://doi.org/10.1109/36.823939, 2000.

9. As the stated by the author, the accuracy of classification was validated by all the training data (see Page 10 Line 6). This is obviously incorrect. I am very confused about the sentence "If the overall accuracy (OA) is lower than 99%, we add 100 patches (50 for ice and 50 for water) from the rest of the training dataset to train the revised model, ……". I'm not sure of your reasons for doing this?

Response: The review is right. We only use a small aspect of samples in the training dataset and the rest for testing. Considering the completeness of the training samples and the problem of overfitting, we only used part of the training sample set for model training, and the remaining samples are used for verification. During the training procedure, we first randomly select 100 samples from each category (sea ice and water) in the training dataset to obtain an initial model, and then use this model on the remaining samples to verify its accuracy. If the accuracy is lower than 99%, 50 samples are added for each class to update the model, and the added training samples are removed from the test samples until the final classification accuracy on the test data is better than 99 %. We repeat the training procedure ten times and find that when the training samples reaches 1000 the accuracy is over 99%. We finally selected 1000 samples for model training (500 for each category), which accounts for 10.25% of the entire training dataset. The table and the corresponding flowchart of the training procedure are listed below and will be included in the next version of the manuscript.

| Step 1 | SAR image selection

One SAR image on each day from June to Sept in 2015-2018 is randomly selected to construct the training data set, finally we get 488 images. |
| --- | --- |
| Step 2 | Training and testing data set construction:

10 patches (samples) for each category (ice and water) with the size of 64*64 pixels are randomly selected from the 488 SAR image using MET Norway ice charts, then we get 9760 patches for constructing the training data set. |
| Step 3 | MSTA-CRF training:

100 patches for each category are selected for training the MSTA-CRF model, and the rest are used as testing samples to decide by the overall accuracy whether the training will be repeated. |

| Step 4 | Testing:

 If the overall accuracy on the testing samples    is larger than 99%, then we get the final MSTA-CRF model, otherwise 100 patches (50 for each category) will be added to retrain the MSTA-CRF model, and the newly selected 100 patches will be removed from the testing samples. |
|--------|---------------------------------------------------------------------------------------------------|
| Step 5 | SAR image classification:

 Repeat step 3 until we train a satisfied model, and the newly trained    model will be used for sea ice and water classification on all the SAR images. |

We also give the flowchart of the training procedure in the following figure.

[Figure]

Minor Comments:

Page 2, line 5: "search-and rescue" --> "search-and-rescue".

Response: corrected.

Page 2, line 6: "ERS-1/-2, RADARSAT-1/-2, Sentinel-1A/-1B" --> "ERS-1/2, RADARSAT-1/2, Sentinel-1A/B".

Response: corrected.

Page 2, line 14: "introduced" --> "have".

Response: corrected.

Page 2, line 15: "channel" --> "polarization". Please replace "channel" with "polarization" in the full text.

Response: We have checked the manuscript and replaced "channel" with "polarization".

Page 2, line 18: "for improved" --> "for improving".

Response: corrected.

Page 2, line 27: here Radarsat-2 is "RS-2", but its abbreviation is "RS2" in line 6.

Response: We have corrected this and finally use RS2 as the abbreviation.

Page 2, line 28: "-3" --> "Sentinel-3".

Response: corrected.

Page 2, line 29: "with low resolution passive microwave form low resolution microwave from AMSR2" reformulate this sentence.

Response: corrected. The operational Ice Service at the Norwegian Meteorological Institute (MET Norway) uses high-resolution SAR data from Sentinel-1, RS2 and COSMO SkyMed, in combination with optical imaging from Sentinel-2 and Sentinel-3, NASA Suomi NPP VIIRS, NOAA AVHRR for visual interpretation, with low-resolution passive microwave from AMSR2 used as a last resort if no other data is available.

Page 2, line 32: "As the backscattering is usually affected by ocean waves propagating into the ice area, …." for thin sea ice, the backscattering coefficient could be affected by wave. But for thick sea ice, the effect of waves on backscattering is very low.

Response: We agree with the reviewer's comment. During the melting season, the backscattering coefficient could be affected by ocean, the sea ice classification accuracy may be affected by using the textual feature based SVM method. Our study is to propose a sea ice classification algorithm for the melting season.

Page 3, line 7: "SVM realize" --> "realizes".

Response: corrected.

Page 3, line 7: "by training the kernel with the transformation into high dimensional space," reformulate this sentence.

Response: corrected. The SVM method uses the kernel function to project the feature into a high-dimensional space.

Page 3, line 9: "Textual feature based neural network methods also shows" --> "show".

Response: corrected.

Page 3, line 10: "Murashkin et al. (2018) use" --> "used".

Response: corrected.

Page 3, line 11: "th MIZ" --> "the MIZ".

Response: corrected.

I stop here with my comments and I think I almost had comments in every single sentence. There are a lot of grammatical issues but also, more seriously, inaccurate statements.

Response: We have gone through the manuscript and revised all the text, and removed ambiguous passages. We have also taken advice from native speakers.